# HEAPr: Hessian-based Efficient atomic expert Pruning in Output Space

**Ke Li**[1]**, Zheng Yang**[2]**, Zhongbin Zhou**[3]**, Feng Xue**[4]**, Zhonglin Jiang**[4]**, Wenxiao Wang**[1][*]

[1] School of Software Technology, Zhejiang University    [2] FABU Inc.
[3] Hangzhou Kuaidi Science and Technology Co., Ltd.
[4] Geely Automobile Research Institute (Ningbo) Co., Ltd

{like2248,wenxiaowang}@zju.edu.cn

## ABSTRACT

Mixture-of-Experts (MoE) architectures in large language models (LLMs) deliver exceptional performance and reduced inference costs compared to dense LLMs. However, their large parameter counts result in prohibitive memory requirements, limiting practical deployment. While existing pruning methods primarily focus on expert-level pruning, this coarse granularity often leads to substantial accuracy degradation. In this work, we introduce HEAPr, a novel pruning algorithm that decomposes experts into smaller, indivisible atomic experts, enabling more precise and flexible atomic expert pruning. To measure the importance of each atomic expert, we leverage second-order information based on principles similar to the Optimal Brain Surgeon theory. To address the computational and storage challenges posed by second-order information, HEAPr exploits the inherent properties of atomic experts to transform the second-order information from expert parameters into that of atomic expert parameters, and further simplifies it to the second-order information of atomic expert outputs. This approach reduces the space complexity from $\mathcal{O}(d^4)$, where $d$ is the model's dimensionality, to $\mathcal{O}(d^2)$. HEAPr requires only two forward passes and one backward pass on a small calibration set to compute the importance of atomic experts. Extensive experiments on MoE models, including DeepSeek MoE and Qwen MoE family, demonstrate that HEAPr outperforms existing expert-level pruning methods across a wide range of pruning ratios and benchmarks. Specifically, HEAPr achieves nearly lossless compression at pruning ratios of $20\% \sim 25\%$ in most models, while also reducing FLOPs by nearly $20\%$. The code can be found at https://github.com/LLIKKE/HEAPr.

## 1 INTRODUCTION

Mixture-of-Experts (MoE) models have recently emerged as a promising alternative to dense large language models (LLMs), replacing dense feed-forward layers with sparsely activated experts and dynamic routing. This design allows MoE models to match or surpass the performance of dense LLMs while activating only a fraction of parameters during inference (Fedus et al., 2022; Zhu et al., 2024; Liu et al., 2024a), making them particularly attractive for large-scale, concurrent deployment. However, while sparse activation reduces computational cost, it exacerbates memory requirements. For example, DeepSeek-V3 (Liu et al., 2024a) activates only 37B parameters per inference, yet all 671B parameters must still be stored in GPU memory, resulting in prohibitively high deployment costs. Notably, MoE layers typically account for over $97\%$ of total model parameters, and they represent the dominant storage bottleneck. Therefore, compressing MoE layers becomes critical to overcoming inference inefficiency and making deployment feasible in resource-constrained devices.

Model pruning has been widely explored as an effective compression strategy to reduce storage and improve efficiency. Yet a fundamental trade-off persists: fine-grained pruning typically preserves accuracy but yields limited speedups on hardware, whereas coarse-grained pruning directly enables acceleration but often incurs obvious accuracy loss. Within MoE models, parameter sparsification (Xie et al., 2024) faces similar limitations, as hardware inefficiencies constrain its practical benefits.

[*]Corresponding author

Consequently, recent research has shifted toward expert-level pruning, offering more direct gains in both acceleration and memory reduction. Existing expert-level approaches at this level can be broadly divided into expert dropping and expert merging. Expert dropping methods (Lu et al., 2024; Huang et al., 2025) completely remove experts deemed unimportant, but relying solely on calibration to discard entire experts risks losing valuable complementary expertise, consequently often leading to notable performance degradation. Expert merging methods (Li et al., 2024; Chen et al., 2025; Huang et al., 2025) instead aim to consolidate functionally similar experts to more effectively preserve overall model capacity. However, their clustering-based similarity measures are notoriously unstable, and naive

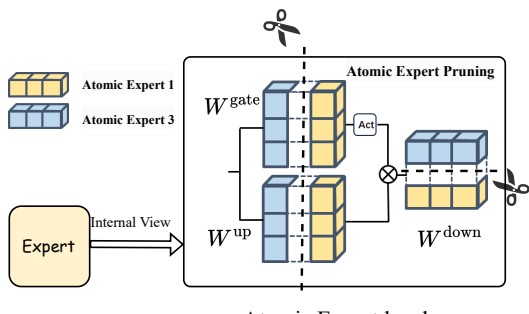

Figure 1: Illustration of atomic expert-level pruning, which removes the $t$-th column from the $W_{gate}$ and $W_{up}$ matrices, and the corresponding $t$-th row from the $W_{down}$ matrix.

merging strategies (e.g., averaging or frequency-based weighting) often introduce destructive parameter conflicts, resulting in suboptimal and inefficient outcomes. To alleviate these critical conflicts, recent decomposition-based approaches (Li et al., 2025c; Gu et al., 2025) represent individual experts as a mixture of shared and specialized components. While this advanced framework helps to preserve model capacity, it still requires computationally expensive decomposition and merging operations, and unfortunately still incurs a non-negligible accuracy loss.

To identify pruning units that are more flexible than expert-level pruning, we introduce the concept of an atomic expert, in which each expert is decomposed into smaller, indivisible units. Concretely, each atomic expert is defined by jointly grouping the relevant columns of $W^{up}$, $W^{gate}$, and the corresponding row of $W^{down}$ (as shown in Figure 1). The output of a full expert can be represented as the sum of outputs from multiple atomic experts. Pruning at this granularity directly removes atomic experts, thereby isolating pruning effects and avoiding interference with remaining components. By eliminating atomic experts that contribute little to final predictions, inference efficiency can be improved and deployment overhead reduced in a more straightforward and essential way.

The key challenge now lies in how to quantify the importance of each atomic expert to overall performance. To tackle this problem, we propose HEAPr, a principled framework for efficient and high-performance atomic expert pruning. Our approach is inspired by the classical Optimal Brain Surgeon (OBS) theory (Hassibi et al., 1993; LeCun et al., 1989), which approximates the effect of weight pruning via a Taylor expansion of the loss function and leverages second-order information to identify parameters with minimal contribution. However, applying OBS to modern deep architectures is computationally prohibitive due to the cost of Hessian estimation, and this is why layer-wise Hessian estimation has become widely adopted (Dong et al., 2017; Frantar & Alistarh, 2022; Frantar et al., 2023). Despite this, the space complexity of Hessian estimation at the expert level remains $\mathcal{O}\left((3d_{\text{model}} \cdot d_{\text{inter}})^2\right)$[1], which is still unacceptable. Therefore, we propose two optimizations to improve Hessian matrix computation. First, by decomposing experts into atomic experts, we demonstrate that the second-order derivatives of parameters between different atomic experts are zero. This observation allows us to significantly reduce the space complexity of the Hessian matrix, lowering it to $\mathcal{O}\left((3d_{\text{model}})^2 \cdot d_{\text{inter}}\right)$. Second, we further optimize the Hessian matrix by shifting the pruning constraints analysis from the parameter space of atomic experts to their output space. This shift enables us to leverage the Fisher information matrix, which is theoretically equivalent to the expected Hessian but significantly more efficient to compute (Bishop & Nasrabadi, 2006; Singh & Alistarh, 2020), and by combining this with a Taylor expansion of the atomic expert function, we can estimate each atomic expert's contribution to the final loss. This further reduces the Hessian complexity to $\mathcal{O}(d_{\text{model}}^2)$ for each expert, ensuring high efficiency in both computation and storage. HEAPr is not only tractable but also highly efficient: all atomic expert importance can be computed with just two forward passes and one backward pass on a small calibration set. We evaluated HEAPr on seven zero-shot tasks, achieving **nearly lossless** pruning with $20\%$ pruning on DeepSeekMoE-16B-Base, $25\%$ pruning on Qwen1.5-MoE-A2.7B-Chat, and $40\%$ pruning on Qwen2-57B-A14B.

---

[1] $d_{\text{inter}}$ is the intermediate dimension after the $W_{\text{up}}$ transformation, and $d_{\text{model}}$ is the hidden size of the model.

Additionally, on the latest Qwen3-30B-A3B model, the average accuracy only drops by $0.03$ at a $25\%$ pruning ratio. Overall, our contributions are summarized as follows:

- We introduce a second-order approximation scheme for atomic expert pruning in MoE models, which transforms the second-order information from expert parameters into that of atomic expert parameters, and further simplifies it to the second-order information of atomic expert outputs. This approach reduces the space complexity of second-order information from $\mathcal{O}\left((3d_{\mathrm{model}} \cdot d_{\mathrm{inter}})^2\right)$ to $\mathcal{O}(d_{\mathrm{model}}^2)$.

- Building on this efficient scheme, we propose HEAPr, a highly efficient and scalable pruning algorithm that accurately estimates the importance of all atomic experts with just two forward passes and one backward pass on a small calibration set.

- We conduct extensive experiments on DeepSeekMoE-16B-Base, Qwen1.5-MoE-A2.7B-Chat, Qwen2-57B-A14B, and Qwen3-30B-A3B across diverse benchmarks. HEAPr outperforms current SOTA methods and achieves nearly lossless compression at pruning ratios of $20\%$–$25\%$ in most models, while also reducing FLOPs by nearly $20\%$.

## 2 RELATED WORKS AND PRELIMINARY

**Mixture of Experts Compression.** Model compression for MoE architectures has recently attracted growing attention due to the remarkable performance of MoE models. MoE-Pruner (Xie et al., 2024) performs weights sparsification based on activation magnitude, weight magnitude, and router importance, yet its acceleration is hardware-dependent and relies on distillation to recover accuracy. Expert-level pruning has been more extensively explored due to its hardware-friendly acceleration. NAEE (Lu et al., 2024) selects a subset of experts to minimize calibration error, but this can lead to overfitting and the loss of specialized knowledge. Similarly, MoE-I$^2$ (Yang et al., 2024) combines expert pruning with low-rank decomposition, yet requires additional fine-tuning for recovery. To alleviate such issues, expert merging methods aim to retain similar experts rather than discarding them. MC-SMoE (Li et al., 2024) merges experts by clustering based on routing policies, and HC-MoE (Chen et al., 2025) does so by grouping experts with similar outputs. However, limited expert similarity makes merging prone to parameter conflicts. EEP (Liu et al., 2024b) uses gradient-free evolutionary search to combine expert dropping and expert merging, cutting SMoE experts and active experts while maintaining or improving downstream performance. To further exploit redundancy, $D^2$-MoE (Gu et al., 2025) constructs a shared expert via weighted combinations and compresses residuals through low-rank decomposition, while Sub-MoE (Li et al., 2025a) applies SVD to extract a shared subspace across experts, both of which require computationally expensive decomposition and merging operations. We decompose the expert into atomic experts and propose HEAPr, a method that measures importance by utilizing a second-order approximation to assess the importance of atomic experts. This approach enables more flexible pruning units and provides an efficient highly algorithm, preserving model performance while eliminating the need for retraining.

**Optimal Brain Surgeon in Pruning.** The OBS framework (Hassibi et al., 1993; LeCun et al., 1989) approaches pruning as an optimization problem, aiming to minimize the increase in the loss function when a parameter is removed. Consider a model that has already been trained and converged, with parameters $\theta$ and a corresponding loss $\ell(\theta)$. We can analyze the effect of perturbing the parameters by analyzing the second-order Taylor expansion of the loss function around $\theta$. Specifically, the change in the loss $\Delta\ell$ when perturbing the parameters by $\delta\theta$ is given by the following:

$$\Delta\ell = \ell(\theta + \delta\theta) - \ell(\theta) = \nabla\ell(\theta)^\top \delta\theta + \frac{1}{2}\delta\theta^\top \boldsymbol{H}\delta\theta + O(\|\delta\theta\|^3), \tag{1}$$

where $\boldsymbol{H}$ is the Hessian matrix of second derivatives of the loss with respect to the model parameters. Since the model has already converged to a local minimum of the loss function, the first-order term can be removed ($\nabla\ell(\theta) = \boldsymbol{0}$), and the higher-order terms can be ignored for small perturbations.

For pruning, the constraint is $\theta_q + \delta\theta_q = 0$ for the target, leading to the optimization problem as:

$$\min_{\delta\theta_q} \quad \frac{1}{2}\delta\theta^\top \boldsymbol{H}\delta\theta, \quad \text{s.t. } \delta\theta_q + \theta_q = 0, \tag{2}$$

where $q$ denotes the index of the pruned parameter. Solving this optimization problem yields the minimal increase in loss from pruning parameter $\theta_q$, which is $\Delta\ell = \frac{1}{2}\frac{\theta_q^2}{[\boldsymbol{H}^{-1}]_{qq}}$.

Directly computing the full Hessian in deep neural networks is practically infeasible. Existing OBS methods to adopt significant approximations. For instance, K-FAC approximation (Martens & Grosse, 2015) provides an efficient approximation of second-order information and Hessians are computed layer-wise to guide pruning (Dong et al., 2017; Frantar & Alistarh, 2022; Frantar et al., 2023). Previous work (Singh & Alistarh, 2020) shows that the Fisher information matrix serves as a reliable Hessian estimate and allows for more efficient computation. Some apply OBS to structured pruning (Yu et al., 2022), but these efforts are limited as they consider only the trace of the Hessian.

## 3 METHOD

### 3.1 ATOMIC EXPERT IN MIXTURE-OF-EXPERTS.

The MoE architecture has been widely adopted in LLMs as a replacement for the dense feed-forward network layer, which effectively increases the model capacity while reducing the number of activated parameters. Formally, given an input token representation $\mathbf{x} \in \mathbb{R}^{d_{\text{model}}}$, the output of the MoE layer with $N_{\text{exp}}$ experts is defined as:

$$\mathbf{y} = \sum_{i=1}^{\kappa} g_i(\mathbf{x})\, E_i(\mathbf{x}), \qquad \mathbf{g}(\mathbf{x}) = \big(g_1(\mathbf{x}), \ldots, g_\kappa(\mathbf{x})\big) = \text{Top-}\kappa\big(\boldsymbol{W}^{\text{gate}}\mathbf{x}\big) \in \mathbb{R}^\kappa, \tag{3}$$

where $\boldsymbol{W}^{\text{gate}} \in \mathbb{R}^{N_{\text{exp}} \times d_{\text{model}}}$ produces router scores and $\text{Top-}\kappa(\cdot)$ denotes the router function that selects the top-$\kappa$ experts. Each expert $E_i(\cdot)$ is a gated feed-forward block:

$$E_i(\mathbf{x}) = \boldsymbol{W}_i^{\text{down}}\big[\, \text{SiLU}\big(\boldsymbol{W}_i^{\text{gate}}\mathbf{x}\big) \odot \big(\boldsymbol{W}_i^{\text{up}}\mathbf{x}\big)\big], \tag{4}$$

where $\boldsymbol{W}_i^{\text{up}}, \boldsymbol{W}_i^{\text{gate}} \in \mathbb{R}^{d_{\text{inter}} \times d_{\text{model}}}$, $\boldsymbol{W}_i^{\text{down}} \in \mathbb{R}^{d_{\text{model}} \times d_{\text{inter}}}$, $\odot$ denotes the Hadamard product, and $\text{SiLU}(\cdot)$ is the SiLU activation. Within each expert, computations can be decomposed into atomic experts. Let $\mathbf{w}_{i,j}^{\text{up}}$ and $\mathbf{w}_{i,j}^{\text{gate}}$ denote the $j$-th rows of $\boldsymbol{W}_i^{\text{up}}$ and $\boldsymbol{W}_i^{\text{gate}}$, respectively, and let $\mathbf{w}_{i,j}^{\text{down}}$ denote the $j$-th column of $\boldsymbol{W}_i^{\text{down}}$. Then the $j$-th atomic expert of the $i$-th expert is

$$\mathbf{e}_i^{(j)}(\mathbf{x}) = \mathbf{w}_{i,j}^{\text{down}}\left[\, \text{SiLU}\big(\mathbf{w}_{i,j}^{\text{gate}}\mathbf{x}\big) \cdot \big(\mathbf{w}_{i,j}^{\text{up}}\mathbf{x}\big)\right] \in \mathbb{R}^{d_{\text{model}}}, \tag{5}$$

where $\mathbf{w}_{i,j}^{\text{up}}, \mathbf{w}_{i,j}^{\text{gate}} \in \mathbb{R}^{1 \times d_{\text{model}}}$ and $\mathbf{w}_{i,j}^{\text{down}} \in \mathbb{R}^{d_{\text{model}} \times 1}$. Consequently, each expert is a linear combination of its atomic experts:

$$E_i(\mathbf{x}) = \sum_{j=1}^{d_{\text{inter}}} \mathbf{e}_i^{(j)}(\mathbf{x}). \tag{6}$$

In this framework, each expert $E_i(\cdot)$ can be viewed as a linear combination of its atomic experts. This decomposition allows pruning at the atomic expert level without compromising the other atomic expert structure, leading to both computational acceleration and deployment efficiency directly.

### 3.2 ATOMIC EXPERT IMPORTANCE ANALYSIS IN THE OUTPUT SPACE

**Importance of Atomic Experts.** As discussed in Section 2, the OBS theory provides an excellent framework for analyzing the impact of parameter pruning on model performance. However, its major limitation is the large Hessian matrix, even when only computed layer-wise. In the case of MoE, directly applying OBS at the expert level is still infeasible, as it requires constructing an exceedingly large Hessian with space complexity of $\mathcal{O}\big((3d_{\text{model}} \cdot d_{\text{inter}})^2\big)$ per expert, leading to prohibitive computation and storage costs. Fortunately, by decomposing the expert into smaller atomic experts, a property is revealed: the parameters of different atomic experts are decoupled, *i.e.*,

$$\frac{\partial^2 E(\mathbf{x})}{\partial \Theta^{(i)}\, \partial \Theta^{(j)}} = \frac{\partial^2 \mathbf{e}^{(i)}(\mathbf{x})}{\partial \Theta^{(i)}\, \partial \Theta^{(j)}} = 0, \quad \forall i \neq j \tag{7}$$

where $\Theta^{(i)} \in \mathbb{R}^{3d_{\text{model}}}$ represents the parameters of the $i$-th atomic expert. This means that the cross-Hessians between different atomic experts are zero, which provides a valuable and simplifying property that allows us to focus exclusively on the Hessian of each individual atomic expert with

respect to its own specific parameters. Based on this observation, the second-order Taylor expansion of the change in the loss function with respect to each expert's parameters can be expressed as:

$$\Delta\ell \approx \frac{1}{2}\delta\Theta^T \boldsymbol{H}\delta\Theta = \frac{1}{2}\sum_{i=1}^{d_{\text{inter}}}(\delta\Theta^{(i)})^T \boldsymbol{H}^{(i)}\delta\Theta^{(i)} \tag{8}$$

here, $\Theta \in \mathbb{R}^{3d_{\text{model}}\cdot d_{\text{inter}}}$ denotes the parameters of a given expert, and $\boldsymbol{H}$ is the corresponding Hessian matrix with space complexity $\mathcal{O}\big((3d_{\text{model}}\cdot d_{\text{inter}})^2\big)$. And each $\boldsymbol{H}^{(i)}$ represents the Hessian for the $i$-th atomic expert. This decomposition leads to a significant reduction in the complexity of summing over the Hessians $\sum_{i=1}^{d_{\text{inter}}}\boldsymbol{H}^{(i)}$, which is reduced to $\mathcal{O}\big((3d_{\text{model}})^2 \cdot d_{\text{inter}}\big)$.

However, the resulting Hessian matrix computation remains unacceptable due to its high computational and storage cost. To further alleviate the bottleneck, we introduce a second optimization that reformulates the pruning constraint. The original parameter-space constraint (equation 2) implies that the atomic expert's output $\mathbf{e}_{\mathcal{P}}(\mathbf{x};\Theta_{\mathcal{P}} + \delta\Theta_{\mathcal{P}})$, where $\Theta_{\mathcal{P}} \in \mathbb{R}^{3d_{\text{model}}}$ denotes the parameters of the atomic expert to be pruned, would be zero for every possible input $\mathbf{x}$. Although theoretically sound, enforcing such a universal constraint is computationally infeasible. This motivates a more targeted reformulation: for a specific token $\mathbf{x}$, what is the minimum loss increase $\Delta\ell(\mathbf{x})$ required to force the expert's output to zero? To make this question concrete, we impose the per-token constraint $\mathbf{e}_{\mathcal{P}}(\mathbf{x};\Theta_{\mathcal{P}} + \delta\Theta_{\mathcal{P}}) = 0$, treating $\mathbf{x}$ as given. Since the atomic expert functions are not optimized with respect to the parameters $\Theta_{\mathcal{P}}$ through gradient descent, applying a Taylor expansion of the atomic expert functions around $\Theta_{\mathcal{P}}$ results in the first-order term dominating, yielding:

$$\mathbf{e}_{\mathcal{P}}(\mathbf{x};\Theta_{\mathcal{P}} + \delta\Theta_{\mathcal{P}}) \approx \mathbf{e}_{\mathcal{P}}(\mathbf{x};\Theta_{\mathcal{P}}) + \boldsymbol{J}_{\mathcal{P}}\delta\Theta_{\mathcal{P}} = \mathbf{0}, \tag{9}$$

where $\boldsymbol{J}_{\mathcal{P}} \in \mathbb{R}^{d_{\text{model}}\times 3d_{\text{model}}}$ denotes the Jacobian of $\mathbf{e}_{\mathcal{P}}(\mathbf{x};\Theta_{\mathcal{P}})$. This leads to the following problem:

$$\min_{\delta\Theta_{\mathcal{P}}} \frac{1}{2}\sum_{i=1}^{d_{\text{inter}}}(\delta\Theta^{(i)})^T \boldsymbol{H}^{(i)}\delta\Theta^{(i)} \quad \text{s.t.} \quad \boldsymbol{J}_{\mathcal{P}}\,\delta\Theta_{\mathcal{P}} + \mathbf{e}_{\mathcal{P}} = \mathbf{0}. \tag{10}$$

To solve the problem in equation 10 , we consider the LLMs trained with a negative log-likelihood loss $\ell$ (e.g., cross-entropy loss). In this setting, the Fisher Information Matrix $\boldsymbol{F}$ is equivalent to the expected Hessian (Bishop & Nasrabadi, 2006), providing a computationally efficient alternative:

$$\mathbb{E}\left[\boldsymbol{H}\right] = \boldsymbol{F} = \mathbb{E}\big[(\nabla_{\Theta}\ell)(\nabla_{\Theta}\ell)^T\big], \tag{11}$$

where $\ell$ is the sample-wise loss. Previous work (Singh & Alistarh, 2020) has shown that for well-converged neural networks, a few hundred representative samples are already sufficiently reliable to estimate $\mathbb{E}\left[\boldsymbol{H}\right]$. Expanding the gradient of $\ell$ with respect to the parameters gives $\nabla_{\Theta_{\mathcal{P}}}\ell = \boldsymbol{J}_{\mathcal{P}}^{\top}\mathbf{g}_{\mathcal{P}}$, where $\mathbf{g}_{\mathcal{P}} \in \mathbb{R}^{d_{\text{model}}}$ is the gradient of the loss with respect to the pruned atomic expert output $\mathbf{e}_{\mathcal{P}}$. Substituting this expression into the objective equation 10 yields the expected loss increase when pruning the atomic expert $\mathbf{e}_{\mathcal{P}}$, with $\delta\Theta^{(i)} = \mathbf{0}$ for all atomic experts not pruned:

$$\tfrac{1}{2}\delta\Theta_{\mathcal{P}}^T\mathbb{E}\left[\boldsymbol{H}_{\mathcal{P}}\right]\delta\Theta_{\mathcal{P}} \approx \mathbb{E}_{\mathbf{x}\sim D}\left[\tfrac{1}{2}\mathbf{e}_{\mathcal{P}}^{\top}\mathbb{E}[\mathbf{g}_{\mathcal{P}}\mathbf{g}_{\mathcal{P}}^{\top}]\mathbf{e}_{\mathcal{P}}\right]. \tag{12}$$

This leads us to define the **Importance** of the atomic expert $\mathbf{e}_{\mathcal{P}}$ as

$$s = \Delta\ell \approx \mathbb{E}_{\mathbf{x}\sim D}\left[\tfrac{1}{2}\mathbf{e}_{\mathcal{P}}^{\top}\mathbb{E}[\mathbf{g}_{\mathcal{P}}\mathbf{g}_{\mathcal{P}}^{\top}]\mathbf{e}_{\mathcal{P}}\right], \tag{13}$$

where a smaller $s$ indicates that the corresponding atomic expert has less impact on the overall model loss and should be pruned with higher priority. The detailed derivation is provided in Appendix A.

At this point, we have shifted the analysis from the parameter space of atomic experts to their output space, further reducing both computational and storage requirements. Next, we introduce a remarkable property of atomic expert outputs: the outputs of atomic experts within the same expert share identical gradients, *i.e.*,

$$\frac{\partial\ell}{\partial\mathbf{e}^{(i)}(\mathbf{x})} = \frac{\partial\ell}{\partial E(\mathbf{x})}, \quad \forall i \in \{1,\dots,d_{\text{inter}}\}, \ \mathbf{e}^{(i)} \in E. \tag{14}$$

This property allows us to further significantly reduce storage requirements. Instead of maintaining separate gradient covariance matrices for each atomic expert, we only need to store a single matrix per expert. As a result, the space complexity for computing the importance of the atomic expert within the same expert is drastically reduced to $\mathcal{O}(d_{\text{model}}^2)$, enabling efficient storage management.

**Global Ranking of Atomic Experts.** The metric of each atomic expert's importance has been introduced by equation 13. Next, an important question is how to rank the importance of the atomic experts. Consider that our importance metric evaluates experts based on their overall contribution to the model's change in the loss function (as shown in equation 2), it provides a natural basis for global ranking. This allows us to effectively compare experts across layers consistently, ensuring that pruning decisions are made based on the entire model's behavior rather than isolated layer-wise.

### 3.3 HEAPR ALGORITHM

Building on the above analysis, we propose HEAPr, a pruning strategy for MoE feedforward layers that ranks the importance of atomic experts (as defined in equation 13) and removes those with negligible contribution to the overall loss. To effectively compute the importance of atomic experts, we leverage a small but representative calibration set $\mathcal{D}$ and estimate importance in two stages.

**1. Shared Gradient Covariance Estimation.** For a given expert $E_i$, the gradients of the loss with respect to all its constituent atomic experts' output are identical. Therefore, rather than performing redundant computations, we execute a single backward pass to obtain the gradient for the expert's output, $\mathbf{g}_{E_i} = \partial \ell / \partial E_i$. This shared gradient is used to compute a gradient covariance matrix $\bar{\mathbf{G}}_i$ for all atomic experts belonging to $E_i$, accumulated over the subset of tokens $\mathcal{T}_i \subseteq \mathcal{D}$ routed to $E_i$:

$$\bar{\mathbf{G}}_i = \frac{1}{|\mathcal{T}_i|} \sum_{\mathbf{x} \in \mathcal{T}_i} \mathbf{g}_{E_i}(\mathbf{x}) \mathbf{g}_{E_i}(\mathbf{x})^\top. \tag{15}$$

**2. Importance Computation.** Subsequently, during a forward pass, we compute the importance for each individual atomic expert $\mathbf{e}_k$. Although the gradient covariance matrix $\bar{\mathbf{G}}_i$ is shared among all atomic experts within same expert $E_i$, the output of each atomic expert, $\mathbf{e}_k(\mathbf{x})$, remains unique. This difference in output allows us to distinguish their individual contributions. The importance of an atomic expert $\mathbf{e}_k$ (where $\mathbf{e}_k \in E_i$) is calculated by averaging over the tokens it processes:

$$\bar{s}_k = \frac{1}{|\mathcal{T}_i|} \sum_{\mathbf{x} \in \mathcal{T}_i} \frac{1}{2} \mathbf{e}_k(\mathbf{x})^\top \bar{\mathbf{G}}_i \mathbf{e}_k(\mathbf{x}). \tag{16}$$

This approach relies solely on standard forward and backward computations, making it both exceptionally time- and memory-efficient. The space complexity of each gradient covariance matrix is only $\mathcal{O}(d_{\text{model}}^2)$, significantly alleviating the storage bottleneck. After computing the importance $\bar{s}_k$ across all atomic experts in the model, we perform a global ranking and prune the lowest $r\%$ of experts across all MoE layers. The complete and optimized procedure is summarized in Algorithm 1.

---

**Algorithm 1** HEAPr: Hessian-based Efficient Atomic Expert Pruning in Output Space

---

**Require:** MoE model $f_\theta$, calibration set $\mathcal{D}$, pruning ratio $r$
**Ensure:** Pruned model $f_{\theta'}$
1: **for** each expert $E_i$ **do**                                   ▷ Stage 1: Gradient Covariance Estimation
2:       Collect routed tokens $\mathcal{T}_i$
3:       Compute shared gradient $\mathbf{g}_{E_i} = \frac{\partial \ell}{\partial E_i}$
4:       Compute $\bar{\mathbf{G}}_i = \frac{1}{|\mathcal{T}_i|} \sum_{\mathbf{x} \in \mathcal{T}_i} \mathbf{g}_{E_i}(\mathbf{x}) \mathbf{g}_{E_i}(\mathbf{x})^\top$            ▷ Space complexity $\mathcal{O}(d^2)$
5: **end for**
6: **for** each atomic expert $\mathbf{e}_k$ in $E_i$ **do**                   ▷ Stage 2: Importance Computation
7:       Compute $\bar{s}_k = \frac{1}{|\mathcal{T}_i|} \sum_{\mathbf{x} \in \mathcal{T}_i} \frac{1}{2} \mathbf{e}_k(\mathbf{x})^\top \bar{\mathbf{G}}_i \mathbf{e}_k(\mathbf{x})$
8: **end for**
9: Global rank $\{\bar{s}_k\}$ and prune lowest $r\%$ across all experts
10: **return** Pruned model $f_{\theta'}$

---

## 4 EXPERIMENTS

### 4.1 EXPERIMENTAL SETUP

**Models and Setup.** We evaluate our approach on a broad spectrum of model architectures and scales to assess its generality and effectiveness, including DeepSeekMoE-16B-Base (Dai et al.,

Table 1: Performance of HEAPr with DeepSeekMoE-16B-Base, Qwen1.5-MoE-A2.7B-Chat, Qwen2-57B-A14B and Qwen3-30B-A3B on seven zero-shot tasks, reported in terms of accuracy. The results marked with * are obtained from the official implementation.

| Ratio | Method | Wiki↓ | PTB↓ | Openb. | ARC_e | WinoG. | HellaS. | ARC_c | PIQA | MathQA | Avg.↑ |
|---|---|---|---|---|---|---|---|---|---|---|---|
| | | | | **DeepSeekMoE-16B-Base** | | | | | | | |
| 0% | Original | 6.38 | 9.47 | 0.32 | 0.76 | 0.71 | 0.58 | 0.45 | 0.79 | 0.32 | 0.56 |
| 20% | NAEE | 9.44 | 15.02 | **0.32** | 0.71 | 0.66 | 0.55 | 0.40 | 0.77 | 0.29 | 0.53 |
| | MoE-I$^2$ | 7.69 | 11.59 | 0.26 | 0.71 | 0.68 | 0.49 | 0.38 | 0.73 | 0.29 | 0.50 |
| | MoE-SVD | 6.92 | 10.48 | 0.31 | 0.75 | 0.70 | 0.53 | 0.42 | 0.76 | 0.31 | 0.54 |
| | $D^2$-MoE | 6.84 | 11.10 | 0.30 | 0.74 | 0.69 | 0.55 | 0.41 | 0.76 | 0.31 | 0.54 |
| | **HEAPr** | **6.54** | **9.88** | **0.32** | **0.76** | **0.71** | **0.57** | **0.45** | **0.79** | **0.32** | **0.56** |
| 40% | NAEE | 8.55 | 14.47 | 0.23 | 0.67 | 0.67 | 0.41 | 0.32 | 0.69 | 0.26 | 0.46 |
| | MoE-I$^2$ | 9.73 | 15.75 | 0.23 | 0.64 | 0.66 | 0.41 | 0.31 | 0.68 | 0.26 | 0.45 |
| | $D^2$-MoE | 7.93 | 14.07 | 0.26 | 0.69 | 0.65 | 0.45 | 0.36 | 0.72 | 0.28 | 0.49 |
| | **HEAPr** | **6.80** | **10.86** | **0.30** | **0.74** | **0.69** | **0.52** | **0.41** | **0.76** | **0.30** | **0.53** |
| | | | | **Qwen1.5-MoE-A2.7B-Chat** | | | | | | | |
| 0% | Original | 8.12 | 12.97 | 0.31 | 0.70 | 0.66 | 0.59 | 0.40 | 0.79 | 0.35 | 0.54 |
| 25% | MC-SMoE | 12.76 | 17.45 | 0.25 | 0.65 | 0.65 | 0.53 | 0.37 | - | - | - |
| | HC-SMoE | 11.62 | 16.39 | 0.27 | 0.66 | 0.63 | 0.55 | 0.35 | **0.76*** | 0.29* | 0.50 |
| | Sub-MoE | 9.48 | 14.84 | 0.30 | **0.69** | 0.66 | **0.56** | 0.37 | - | - | - |
| | **HEAPr** | **8.31** | **14.12** | **0.32** | **0.69** | **0.67** | **0.56** | **0.38** | 0.76 | **0.35** | **0.53** |
| 50% | MC-SMoE | 5e2 | 1e3 | 0.18 | 0.33 | 0.52 | 0.29 | 0.19 | - | - | - |
| | HC-SMoE | 25.50 | 38.18 | 0.23 | 0.61 | **0.65** | **0.47** | **0.35** | 0.58* | 0.23* | 0.45 |
| | Sub-MoE | 17.51 | 29.00 | 0.25 | 0.58 | 0.58 | 0.46 | 0.25 | - | - | - |
| | **HEAPr** | **9.24** | **17.58** | **0.27** | **0.64** | 0.64 | 0.46 | 0.33 | **0.71** | **0.33** | **0.48** |
| | | | | **Qwen3-30B-A3B** | | | | | | | |
| 0% | Original | 8.64 | 15.40 | 0.34 | 0.79 | 0.71 | 0.60 | 0.54 | 0.79 | 0.59 | 0.62 |
| 25% | HC-SMoE | 18.86 | 31.11 | 0.22 | 0.64 | 0.61 | 0.40 | 0.35 | 0.59* | 0.41* | 0.46 |
| | Sub-MoE | 13.59 | 23.48 | 0.25 | 0.70 | 0.66 | 0.47 | 0.44 | - | - | - |
| | **HEAPr** | **9.10** | **16.80** | **0.33** | **0.76** | **0.70** | **0.55** | **0.49** | **0.78** | **0.50** | **0.59** |
| 50% | HC-SMoE | 72.33 | 162.99 | 0.13 | 0.44 | 0.50 | 0.29 | 0.23 | 0.44* | 0.32* | 0.34 |
| | Sub-MoE | 21.05 | 43.19 | 0.23 | **0.68** | 0.63 | **0.41** | 0.40 | - | - | - |
| | **HEAPr** | **11.22** | **26.29** | **0.25** | 0.67 | **0.63** | 0.38 | **0.41** | **0.67** | **0.36** | **0.48** |
| | | | | **Qwen2-57B-A14B** | | | | | | | |
| 0% | Original | 5.12 | 9.18 | 0.33 | 0.75 | 0.74 | 0.63 | 0.46 | 0.81 | 0.39 | 0.59 |
| 40% | NAEE | 6.81 | 11.34 | 0.31 | 0.73 | 0.73 | 0.55 | 0.46 | 0.76 | 0.36 | 0.55 |
| | MoE-I$^2$ | 24.90 | 77.05 | 0.26 | 0.70 | 0.46 | 0.71 | 0.41 | 0.75 | 0.30 | 0.51 |
| | $D^2$-MoE | 8.19 | 11.23 | **0.33** | **0.75** | **0.75** | 0.61 | 0.45 | 0.79 | 0.36 | 0.58 |
| | **HEAPr** | **5.92** | **9.34** | **0.33** | **0.75** | 0.74 | **0.63** | **0.46** | **0.81** | **0.39** | **0.59** |

2024), Qwen1.5-MoE-A2.7B-Chat (Team, 2024a), Qwen2-57B-A14B (Team, 2024b), and Qwen3-30B-A3B. All experiments are calibrated on WikiText-2 using 128 sequences of 2048 tokens (see Appendix B for details). Notably, our method introduces no additional tunable hyperparameters.

**Baselines.** For our comparisons, we evaluate seven recently proposed high-performance compression methods, including expert dropping (NAEE (Lu et al., 2024), MoE-I$^2$ (Yang et al., 2024)), expert merging (MC-SMoE (Li et al., 2024), HC-SMoE (Chen et al., 2025)), and expert decomposition (Sub-MoE (Li et al., 2025a), $D^2$-MoE (Gu et al., 2025), MoE-SVD (Li et al., 2025b)). Baseline data were collected from prior publications, prioritizing original sources, and details are provided in Appendix B. Missing data for open-source implementations were obtained from official code.

**Evaluation.** We report results on seven zero-shot benchmarks using the LM-Evaluation-Harness (version 0.4.7) (Gao et al., 2024), including HellaSwag (Zellers et al., 2019), MathQA (Amini et al., 2019), OpenBookQA (OBQA) (Mihaylov et al., 2018), PIQA (Bisk et al., 2020), WinoGrande (Sakaguchi et al., 2021), ARC-Easy and ARC-Challenge (Boratko et al., 2018). These tasks collectively enable repeated and consistent evaluation of our method across varied domains.

## 4.2 MAIN RESULTS

**Compression Performance.** As shown in Table 1, HEAPr achieves exceptional performance across various MoE models and pruning ratios. Notably, our method delivers near-lossless compression. At pruning ratios of $20\% \sim 25\%$, HEAPr matches the performance of the original models on DeepSeekMoE-16B-Base and Qwen1.5-MoE-A2.7B-Chat. More impressively, on Qwen2-57B-A14B, HEAPr maintains performance almost identical to the original model even at a high $40\%$ compression ratio. In contrast, our method outperforms recent approaches such as Sub-MoE, $D^2$-MoE, and NAEE under the same compression ratio. Furthermore, on the latest Qwen3-30B-A3B model, HEAPr incurs only a minimal performance loss at a $25\%$ pruning ratio, with the average accuracy dropping slightly from $0.62$ to just $0.59$. These results strongly highlight the unique advantage of HEAPr in pruning at the atomic expert level, enabling substantial model efficiency improvements while maintaining and preserving core model performance effectively.

**Compare to CAMERA-P.** In this section, we compare HEAPr with a concurrent related work CAMERA-P (Xu et al., 2025), which evaluates the importance of an atomic expert using the concept of decoding-time energy. Specifically, the importance of the $j$-th atomic expert in the $i$-th expert is given by $\varepsilon_{i,j} = (||\Phi_{i,j}||_2 + \alpha||\Phi_{i,j}||_\infty) \cdot ||\mathbf{w}_{i,j}^{\text{down}}||_2$, where $\Phi_{i,j} = \text{SiLU}\left(\mathbf{w}_{i,j}^{\text{gate}}\mathbf{x}\right) \cdot \left(\mathbf{w}_{i,j}^{\text{up}}\mathbf{x}\right)$. CAMERA-P uses a heuristic approach to measure atomic expert importance based on the output magnitudes on a calibration set. However, this method has two main drawbacks: it is local, neglecting atomic experts' impact on overall model performance and cannot be globally applied for pruning due to varying activation magnitudes across layers. In contrast, our method HEAPr, built upon the OBS framework, leverages the Hessian matrix to assess the impact of atomic experts on the overall model performance. And HEAPr yields a globally consistent importance metric for atomic experts, thereby enabling principled global pruning, as analyzed in Section 3.2. In Table 2, we compare the performance of HEAPr and CAMERA-P on DeepSeekMoE-16B-Base. Since CAMERA-P has not released its open-source implementation, we evaluate HEAPr using the `acc_norm` as reported in the original paper. At a $20\%$ pruning ratio, HEAPr outperforms CAMERA-P by an average of $1.2$ in accuracy. Even when applying the same layer-wise pruning strategy as CAMERA-P, HEAPr still achieves an average accuracy improvement of $0.5$. Notably, at a $40\%$ pruning ratio, the performance gap between the two methods narrows. We attribute this to the reduced redundancy at higher pruning ratio, where the non-essential atomic experts identified by both methods become nearly identical.

**Performance Boundary of Pruning.** Figure 2 reports the performance of HEAPr on DeepSeekMoE-16B-Base using a random 128-sample subset of WikiText-2 with 2048 tokens under different compression ratios, where the ratio denotes the fraction of parameters removed relative to the full model size. For compression ratios below 0.4, the pruned models retain 93% of the baseline accuracy while achieving a 20% reduction in FLOPs. In this regime, the accuracy curve is nearly flat, indicating substantial redundancy among atomic experts and suggesting that HEAPr can effectively identify and remove them. As compression increases further, accuracy degrades gracefully, highlighting a clear trade-off between efficiency and performance. Even at an extreme

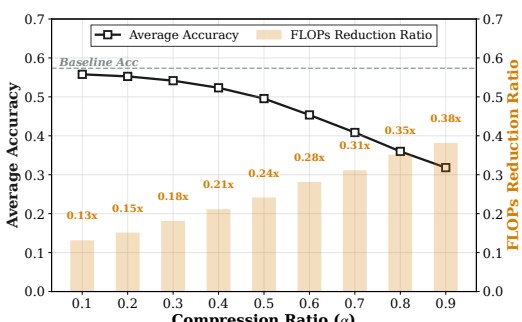

Figure 2: Performance of DeepSeekMoE-16B-Base under varying compression ratios, with corresponding FLOPs saving on WikiText2 data.

compression ratio of 0.9, the model retains about 38% of the baseline accuracy while achieving a 55% reduction in FLOPs. These results demonstrate the robustness of HEAPr under moderate pruning and its effectiveness in enabling aggressive acceleration while retaining reasonable performance.

## 4.3 ABLATIONS

**Global vs. Layer-wise Pruning.** As shown in Table 2, layer-wise pruning (CAMERA-P, HEAPr-L) ranks the importance of atomic experts within each MoE layer and prunes the bottom $r\%$, whereas

Table 2: Comparison of layer-wise pruning (CAMERA-P and HEAPr-L) versus global pruning (HEAPr-G) on DeepSeekMoE-16B-Base and Qwen1.5-MoE-A2.7B-Chat across seven zero-shot tasks, with `acc_norm` reported for DeepSeekMoE-16B-Base and accuracy for others.

| Ratio | Method | Openb. | ARC_e | WinoG. | HellaS. | ARC_c | PIQA | MathQA | Average |
|---|---|---|---|---|---|---|---|---|---|
| | | **DeepSeekMoE-16B-Base** | | | | | | | |
| | CAMERA-P | 44.00 | 71.80 | 70.17 | 75.02 | 45.56 | 78.62 | **31.46** | 59.52 |
| 20% | HEAPr-L | 44.01 | 72.64 | 70.01 | 75.55 | **47.27** | 79.76 | 30.92 | 60.03 |
| | HEAPr-G | **44.80** | **73.73** | **71.43** | **76.57** | 47.01 | **79.82** | 31.42 | **60.68** |
| | CAMERA-P | **43.20** | 70.71 | 68.51 | 69.04 | 42.24 | 75.41 | 29.01 | 56.87 |
| 40% | HEAPr-L | 42.80 | 70.45 | 68.35 | 68.10 | 43.69 | 76.12 | 29.41 | 56.99 |
| | HEAPr-G | 41.40 | **72.05** | **69.06** | **70.79** | **45.05** | **76.39** | **29.85** | **57.80** |
| | | **Qwen1.5-MoE-A2.7B-Chat** | | | | | | | |
| 25% | HEAPr-L | 30.60 | 66.58 | 66.77 | 55.09 | **38.05** | **76.61** | 33.97 | 52.52 |
| | HEAPr-G | **31.80** | **68.60** | **67.22** | **55.67** | 37.56 | 76.39 | **34.87** | **53.59** |
| 50% | HEAPr-L | 27.00 | 63.26 | 64.01 | **47.00** | 33.70 | 69.80 | 32.29 | 48.15 |
| | HEAPr-G | **27.01** | **63.89** | **64.32** | 46.35 | **34.22** | **70.86** | **33.37** | **48.57** |

global pruning (HEAPr-G) ranks the importance of all atomic experts across the entire model. Compared with CAMERA-P, our layer-wise pruning HEAPr-L achieves superior performance, indicating that the atomic expert importance metric, as derived from equation 13, provides a more effective pruning criterion within individual layers. Furthermore, HEAPr-G, by leveraging global pruning and importance scores across all layers, achieves even stronger and more consistent results, validating the global consistency of the atomic expert importance thoroughly analyzed in Section 3.2.

**Impact of Pruning Granularity.** To better demonstrate the importance of atomic expert decomposition, we conduct an ablation study comparing pruning at the atomic expert level and expert level. Based on equation 8, the importance score for an expert computed via equation 13 can be expressed as the sum of the importance scores of its constituent atomic experts. As reported in Table 3, expert-level pruning behaves similarly to Expert Dropping (Lu et al., 2024): the activated experts unchanged after pruning does not lead to noticeable computational speedup. In contrast, pruning at the atomic expert level reduces the dimensionality within each expert, thereby enabling real acceleration. Empirically, atomic-level pruning consistently outperforms expert-level pruning across multiple benchmarks, highlighting its effectiveness and necessity.

**Empirical Correlation of Loss and Atomic Expert Importance** $s_k$**.** In Section 2, following the principles of OBS theory, we define the atomic expert importance score $s_k$ for $\mathbf{e}_{\mathcal{P}}$ based on the expected change in model loss. The goal of this metric is to identify atomic experts whose removal induces the smallest increase in the overall loss. However, because both the OBS formulation and the output-space approximation neglect higher-order terms, an exact numerical match between $s_k$ and the empirical loss change $\Delta\ell$ is not expected. Importantly, pruning ultimately requires a reliable ranking of atomic expert importance rather than an accurate prediction of $\Delta\ell$. To evaluate the ranking quality of $s_k$, we infer the atomic experts on the calibration set and then group them into 10% bins according to their original indices. As shown in Figure 3, the observed loss increase

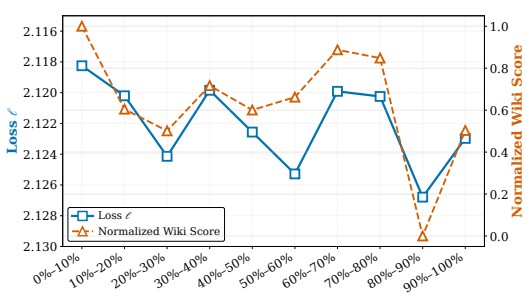

Figure 3: Consistency between atomic expert normalized importance score $s_k$ and the change in loss. The figure plots the actual loss increase $\Delta\ell$ observed upon pruning atomic experts within 10% quantile bins (ordered by original expert index) against the cumulative importance score $s_k$.

$\Delta\ell$ for each bin closely follows the cumulative trend of the corresponding normalized importance scores $s_k$. This result indicates that, despite the approximations involved, the proposed $s_k$ metric

provides a globally consistent and reliable ranking of atomic experts. It effectively identifies atomic experts whose removal causes minimal performance degradation, thereby offering a solid basis for the HEAPr algorithm and supporting the accuracy of our pruning decisions.

Table 3: Comparison of pruning granularities at the expert level and the atomic expert level, where expert importance is computed by summing the importances of its atomic experts, evaluated across seven zero-shot tasks. FLOPs rr. denotes the FLOPs reduction ratio. Results are reported on DeepSeekMoE-16B-Base.

| Ratio | Level | FLOPs rr.↑ | Wiki↓ | Openb. | ARC_e | WinoG. | HellaS. | ARC_c | PIQA | MathQA |
|---|---|---|---|---|---|---|---|---|---|---|
| 20% | Expert | 0% | 6.90 | 31.40 | 75.76 | 71.35 | **57.99** | 44.31 | 78.40 | 30.75 |
| | Atomic Expert | 8% | **6.54** | **31.54** | **75.88** | **71.43** | 57.39 | **44.62** | **79.05** | **31.52** |
| 40% | Expert | 0% | 8.00 | 30.60 | 73.19 | 63.93 | 51.15 | **42.49** | **77.09** | 28.24 |
| | Atomic Expert | 21% | **6.80** | **30.00** | **73.78** | **69.06** | **52.29** | 40.61 | 76.50 | **30.12** |

**Impact of Calibration Data.** Figure 4 shows the average accuracy with error bars over random subsets of the calibration data, indicating that the performance of our HEAPr algorithm is largely unaffected by the choice of calibration data, whether they are WikiText-2 or C4 dataset. This highlights the remarkable robustness and generalizability of our method, as it consistently performs well across different calibration corpora and domains. Furthermore, the table also explores the significant impact of calibration set size on pruning performance. As the number of calibration samples increases, the model's performance improves consistently, indicating that larger calibration sets offer richer statistical coverage, which provides more reliable and informative signals for

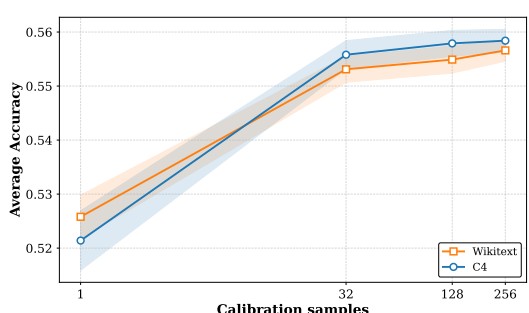

Figure 4: Performance of DeepSeekMoE-16B-Base under a 20% compression ratio, using calibration data randomly sampled from WikiText-2 and C4.

effective compression. These results strongly suggest that our method is not only robust to variations in calibration data but also benefits from the inclusion of additional diverse samples.

## 5  CONCLUSION

In this work, we introduce HEAPr, a novel method that refines expert-level pruning in MoE models by enabling a more flexible and fine-grained pruning strategy at the atomic expert level. Inspired by the principles of the Optimal Brain Surgeon theory, we evaluate the importance of atomic experts using second-order information. By transforming the analysis from the expert parameter space to that of atomic expert parameters, and further shifting it to the atomic expert output space, we significantly reduce the computational and storage bottlenecks associated with the second-order information matrix. HEAPr requires only two forward passes and one backward pass to efficiently compute the importance of atomic experts. Extensive experiments on various modern MoE models demonstrate that HEAPr outperforms state-of-the-art pruning methods, achieving near-lossless pruning with pruning rates of $20\% \sim 25\%$. More importantly, our method provides a much finer-grained perspective on MoE expert pruning, which we hope will contribute to a deeper, more comprehensive understanding of MoE models. Future work will explore large-scale experiments across a wider range of models and investigate the potential of parameter compensation methods after the pruning.

## ACKNOWLEDGMENTS

This work was supported in part by The National Nature Science Foundation of China (Grant NO.: 62303406), in part by Ningbo Key R&D Program (NOs.: 2025Z055, 2025Z035), in part by Yongjiang Talent Introduction Programme (Grant NO.: 2023A-194-G), in part by Zhiyuan Laboratory (NO.: ZYL2024022b).

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

## APPENDIX OVERVIEW

## A  DERIVATION OF THE IMPORTANCE OF ATOMIC EXPERT

We provide a detailed derivation of the importance measure introduced in equation 13.

Let $\Theta$ denote the model parameters of dimension $P$. Let $\mathbf{e}_k(\mathbf{x}) \in \mathbb{R}^d$ be the output of the $k$-th atomic expert for a given input $\mathbf{x}$. We define the Jacobian of the atomic expert output with respect to the parameters as $\boldsymbol{J}_k(\mathbf{x}) = \frac{\partial \mathbf{e}_k(\mathbf{x})}{\partial \Theta} \in \mathbb{R}^{d \times P}$. Let $\mathbf{g}(y) = \nabla_{\mathbf{e}_k} \log P(y|\mathbf{x}, \Theta) \in \mathbb{R}^d$ be the gradient of the log-likelihood with respect to the expert output.

Under the negative log-likelihood loss, the Hessian matrix $H$ can be approximated by the Fisher Information Matrix $\boldsymbol{F}$:

$$\boldsymbol{H} \approx \boldsymbol{F} = \mathbb{E}_{\mathbf{x} \sim P_{\text{data}}} \left[ \mathbb{E}_{y \sim P_{\text{model}}(y|\mathbf{x})} \left[ \nabla_{\Theta} \log P(y|\mathbf{x}, \Theta) \nabla_{\Theta} \log P(y|\mathbf{x}, \Theta)^{\top} \right] \right]. \tag{17}$$

Using the chain rule, the gradient of the log-likelihood with respect to the parameters can be decomposed as:

$$\nabla_{\Theta} \log P(y|\mathbf{x}, \Theta) = \boldsymbol{J}_k(\mathbf{x})^{\top} \mathbf{g}(y). \tag{18}$$

Substituting this into the expression for $\boldsymbol{F}$:

$$\boldsymbol{F} = \mathbb{E}_{\mathbf{x}} \left[ \mathbb{E}_{y|\mathbf{x}} \left[ \boldsymbol{J}_k(\mathbf{x})^{\top} \mathbf{g}(y) \mathbf{g}(y)^{\top} \boldsymbol{J}_k(\mathbf{x}) \right] \right]. \tag{19}$$

Since $\boldsymbol{J}_k(\mathbf{x})$ depends only on $\mathbf{x}$ and is constant with respect to the inner expectation over $y$, we can extract it:

$$\boldsymbol{F} = \mathbb{E}_{\mathbf{x}} \left[ \boldsymbol{J}_k(\mathbf{x})^{\top} \left( \mathbb{E}_{y|\mathbf{x}} \left[ \mathbf{g}(y) \mathbf{g}(y)^{\top} \right] \right) \boldsymbol{J}_k(\mathbf{x}) \right]. \tag{20}$$

We define $\boldsymbol{G}_k(\mathbf{x}) \triangleq \mathbb{E}_{y|\mathbf{x}} \left[ \mathbf{g}(y) \mathbf{g}(y)^{\top} \right]$. Then, the parameter-space FIM becomes:

$$\boldsymbol{F} = \mathbb{E}_{\mathbf{x}} \left[ \boldsymbol{J}_k(\mathbf{x})^{\top} \boldsymbol{G}_k(\mathbf{x}) \boldsymbol{J}_k(\mathbf{x}) \right]. \tag{21}$$

We aim to find a parameter perturbation $\delta\Theta$ that minimizes the change in the overall loss while canceling the output of the $k$-th atomic expert (i.e. $\boldsymbol{J}_k(\mathbf{x})\delta\Theta + \mathbf{e}_k(\mathbf{x}) = 0$). The optimization problem is formulated as:

$$\min_{\delta\Theta} \; \tfrac{1}{2} \delta\Theta^{\top} \boldsymbol{F} \, \delta\Theta \qquad \text{s.t.} \qquad \boldsymbol{J}_k(\mathbf{x}) \, \delta\Theta + \mathbf{e}_k(\mathbf{x}) = 0. \tag{22}$$

Substituting the derived form of $\boldsymbol{F}$:

$$\min_{\delta\Theta} \; \tfrac{1}{2} \delta\Theta^{\top} \mathbb{E}_{\mathbf{x}} \left[ \boldsymbol{J}_k(\mathbf{x})^{\top} \boldsymbol{G}_k(\mathbf{x}) \boldsymbol{J}_k(\mathbf{x}) \right] \delta\Theta. \tag{23}$$

Since $\delta\Theta$ is a global parameter update independent of the specific sample $\mathbf{x}$, we can move it inside the expectation:

$$\min_{\delta\Theta} \ \tfrac{1}{2} \mathbb{E}_{\mathbf{x}}\big[(\boldsymbol{J}_k(\mathbf{x})\delta\Theta)^\top \boldsymbol{G}_k(\mathbf{x})(\boldsymbol{J}_k(\mathbf{x})\delta\Theta)\big]. \tag{24}$$

Using the constraint $\boldsymbol{J}_k(\mathbf{x})\delta\Theta = -\mathbf{e}_k(\mathbf{x})$, we substitute this into the objective function:

$$S_k = \tfrac{1}{2} \mathbb{E}_{\mathbf{x}}\big[\mathbf{e}_k(\mathbf{x})^\top \boldsymbol{G}_k(\mathbf{x})\mathbf{e}_k(\mathbf{x})\big]. \tag{25}$$

## B  DETAIL ANALYSIS OF MAIN RESULTS.

**Calibration Set Sampling Strategy.**  To construct the calibration set, we first load the entire dataset (either WikiText-2 or C4) and concatenate all sentences into a single corpus using "\n\n" as the separator. We then tokenize the full corpus and split the resulting token stream into consecutive samples, each consisting of 2048 tokens. With a fixed random seed (random.seed(0)) for reproducibility, we randomly select 128 such samples to form the calibration set. The 128 samples drawn from WikiText were used to obtain all results reported in Table 1, and the impact of the calibration set is discussed in Section 4.3.

**Details of Baseline Experiments.**  For DeepSeekMoE-16B-Base, the results for NAEE, MoE-I$^2$, and $D^2$-MoE are taken from the paper (Gu et al., 2025), while MoE-SVD results are sourced from its paper (Li et al., 2025b). For Qwen1.5-MoE-A2.7B-Chat, all results are from the paper (Li et al., 2025a); any missing results with available official open-source code were reproduced by us. For Qwen3-30B-A3B, the results for HC-SMoE and Sub-MoE are from the paper (Li et al., 2025a). For Qwen2-57B-A14B, the results for NAEE, MoE-I$^2$, and $D^2$-MoE are taken from the paper (Gu et al., 2025). Table 4 shows the calibration dataset size for various methods.

Table 4: Calibration set sizes for different methods (2048 sqlen).

| Method | NAEE | $D^2$-MoE | Sub-MoE | HEAPr |
|---|---|---|---|---|
| Calibration Set Size | 128 | 512 | 128 | 128 |

## C  ANALYSIS OF RUNTIME SPEEDUP AND MEMORY USAGE

Table 5 summarizes the computational cost and performance of HEAPr compared with competitive baseline methods (NAEE and $D^2$-MoE) on two representative MoE models: DeepSeekMoE-16B-Base and Qwen2-57B-A14B. The table reports the number of calibration samples used for pruning, the theoretical FLOPs (TFLOPs) required for pruning, GPU time cost, peak memory usage.

Table 5: Comparison of computational cost between HEAPr and baseline pruning methods.

| Method | Samples | TFLOPs | GPU Time Cost | Memory |
|---|---|---|---|---|
| DeepSeekMoE-16B-Base | | | | |
| NAEE | 128 | 11 | 2 min | 27GB |
| $D^2$-MoE | 512 | 227 | 30 min | 53GB |
| HEAPr | 128 | 44 | 6 min | 44GB |
| Qwen2-57B-A14B | | | | |
| NAEE | 128 | 32 | 8 min | 60GB |
| $D^2$-MoE | 512 | 1205 | 90 min | 127GB |
| HEAPr | 128 | 123 | 20 min | 91GB |

## D    COMPRESSION RATE ANALYSIS UNDER GLOBAL PRUNING

In this section, we analyze the compression rates across different layers when applying a 25% and 50% global pruning strategy based on the global ranking of atomic experts' importance. As shown in Figure 5 and  6, the compression rate is initially high in the early layers, suggesting that the experts in these layers are less important and can be pruned with minimal impact on the model's performance. As we move deeper into the network, the compression rate decreases, indicating that the experts in these layers are more important to the model's performance. Interestingly, after a certain point, the compression rate starts to increase again in the deepest layers, suggesting that some experts in these layers become redundant, allowing for further pruning without significant loss of model performance. This non-monotonic behavior highlights the varying importance of experts across layers in MoE-based models.

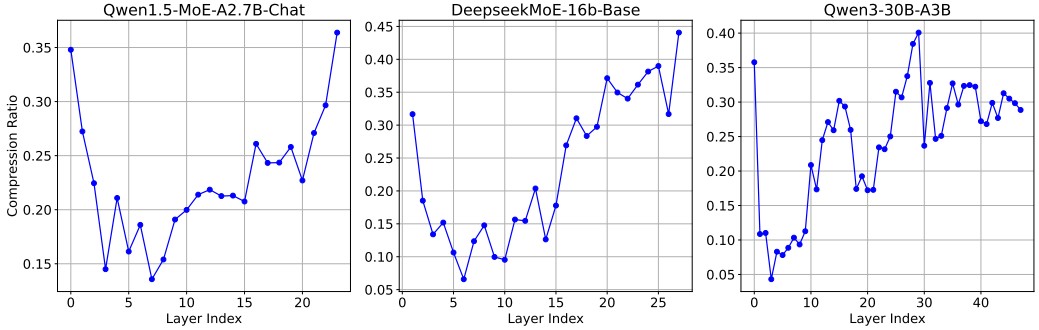

Figure 5: Compression ratios across different layers under 25% global pruning for Qwen1.5-MoE-A2.7B-Chat, DeepSeekMoE-16b-Base, and Qwen3-30B-A3B.

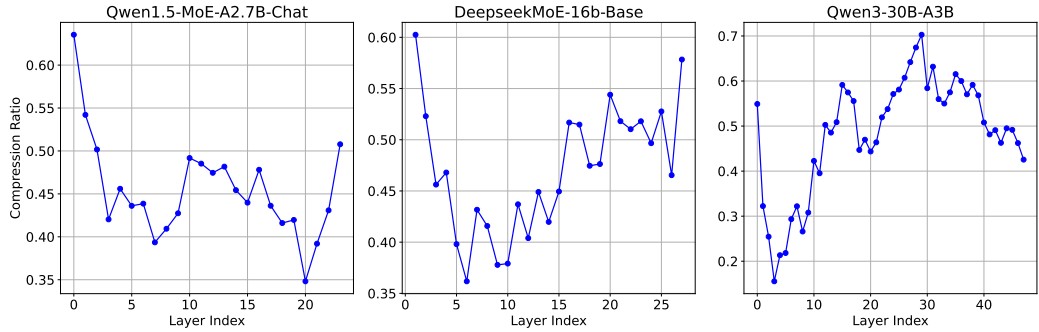

Figure 6: Compression ratios across different layers under 50% global pruning for Qwen1.5-MoE-A2.7B-Chat, DeepSeekMoE-16b-Base, and Qwen3-30B-A3B.

## E    USE OF LLMS

In this work, Large Language Models (LLMs) were primarily utilized for tasks such as text refinement, offering writing suggestions, and improving the overall structure and clarity of the manuscript. It is important to note that LLMs did not contribute to the ideation or development of the methodology section. The authors guarantee that all LLM-generated content was thoroughly reviewed and edited to ensure its accuracy and coherence.

## F    REPRODUCIBILITY STATEMENT

To ensure reproducibility, we have made the code and checkpoints obtained in our computational environment available at https://github.com/LLIKKE/HEAPr. While we have taken every effort to

ensure consistency, results may exhibit slight variations due to the random selection of calibration sets, as well as potential version differences in libraries such as transformers and LM-Evaluation-Harness. These fluctuations are expected and considered acceptable.

## G    ETHICS STATEMENT

This work adheres to ethical guidelines in conducting research and reporting results. We have used publicly available datasets and models, ensuring that our methods comply with their respective terms of use. The research itself aims to enhance existing technologies and does not introduce any ethical concerns. No personal or sensitive data was used in this study, and the methods employed do not raise any known ethical issues.

