# OpenReview forum: "HEAPr: Hessian-based Efficient Atomic Expert Pruning in Output Space"
_ICLR.cc/2026/Conference — ICLR 2026 Poster_

### Official Review · Reviewer_UZ1E · 2025-10-18

**Soundness:** 3
**Presentation:** 3
**Contribution:** 2
**Rating:** 4
**Confidence:** 3

**Summary:**

This paper introduces HEAPr, a novel pruning method for Mixture-of-Experts (MoE) models that decomposes experts into smaller atomic experts and leverages second-order information based on Optimal Brain Surgeon theory to assess their importance. By transforming the analysis from expert parameter space to atomic expert output space, the method reduces computational complexity and enables more precise fine-grained pruning.

**Strengths:**

- The method is well-grounded in Optimal Brain Surgeon theory and provides clear mathematical derivations for adapting second-order information to MoE architectures. The theoretical framework is comprehensive and enhances the method's credibility.
-  The paper effectively addresses the fundamental challenge of applying Optimal Brain Surgeon theory to modern large-scale models by cleverly shifting the pruning analysis from high-dimensional parameter space to a more tractable output space, thereby overcoming the computational difficulties in MoE

**Weaknesses:**

- The experimental evaluation of the proposed HEAPr method is conducted solely on models from the DeepSeekMoE and Qwen families. This narrow scope raises concerns about the method's generalizability to other model families (such as Mixtral or GPT-OSS), and it remains unclear whether the method would maintain its effectiveness across diverse MoE implementations and architectures.
- The evaluation is limited to commonsense reasoning and QA benchmarks, notably omitting crucial domains like code generation, which prevents a comprehensive assessment of the pruned model's full capabilities.
-  The paper emphasizes the efficiency of HEAPr, yet the experiments provide no quantitative data to support this claim. The evaluation is missing a crucial comparison of the computational cost of the pruning process itself when compared to baseline methods.
- The paper presents the "atomic expert" as a key contribution, but acknowledges and compares against concurrent work CAMERA[1], which introduces an identically defined structure called "Micro-Expert." This suggests that the atomic expert pruning granularity may not be original to this work, but rather represents an emerging common approach in the field. Moreover, this pruning approach is conceptually similar to structured pruning within experts, as both involve removing corresponding columns/rows across multiple matrices.

[1] CAMERA: Multi-Matrix Joint Compression for MoE Models via Micro-Expert Redundancy Analysis

**Questions:**

1. How does HEAPr perform on other major MoE model families beyond DeepSeekMoE and Qwen?

2. How does HEAPr perform on other tasks?

3. What are the quantitative comparisons of computational costs between HEAPr and baseline methods?

---

> ### Author Response · Authors · 2025-11-26
>
> Thank you  for your valuable feedback. Your main concerns are: (1) Experiments on more models (Mixtral 8×7B, GPT-oss-20B): we added their pruning results; (2) Additional LLM evaluations: we included perplexity, knowledge (57 tasks) , and code generation; (3) HEAPr efficiency: we compared it with SOTA methods; (4) Novelty: we clarified key differences from concurrent work. Below, we provide point-by-point responses to your comments.
>
> **Q1:** How does HEAPr perform on other major MoE model families beyond DeepSeekMoE and Qwen?
>
> >**W1:** The experimental evaluation of the proposed HEAPr method is conducted solely on models from the DeepSeekMoE and Qwen families. This narrow scope raises concerns about the method's generalizability to other model families (such as Mixtral or GPT-OSS), and it remains unclear whether the method would maintain its effectiveness across diverse MoE implementations and architectures.
>
> **A1:** Thank you for the suggestion. We agree that including more models would further strengthen the paper. Due to space limitations in Table 1, we initially reported only the most recent and widely used models. To address your concern and better demonstrate the generalizability of HEAPr, we now provide results for additional MoE models, including Mixtral-8x7B and GPT-OSS-20B:
>
> | Model        | Method | Pruning Ratio | obqa     | ARC_e    | WinoG    | HellaS   | ARC_C    | PIQA     | MathQA   |
> | ------------ | ------ | ------------- | -------- | -------- | -------- | -------- | -------- | -------- | -------- |
> | Mixtral-8x7B | Origin | 0             | 0.36     | 0.84     | 0.76     | 0.65     | 0.57     | 0.82     | 0.43     |
> |              | NAEE   | 0.2           | 0.32     | 0.76     | 0.72     | 0.58     | **0**.47 | 0.79     | **0.40** |
> |              | HEAPr  | 0.2           | **0.33** | **0.80** | **0.69** | **0.56** | **0.47** | **0.79** | 0.38     |
>
> | Model       | Method | Pruning Ratio | obqa  | ARC_e | WinoG | HellaS | ARC_C | PIQA  | MathQA |
> | ----------- | ------ | ------------- | ----- | ----- | ----- | ------ | ----- | ----- | ------ |
> | gpt-oss-20b | Origin | 0             | 27.70 | 77.48 | 66.61 | 41.60  | 45.22 | 75.57 | 36.45  |
> |             | NAEE   | 0.2           | 23.20 | 70.31 | 63.65 | 34.56  | 35.32 | 70.21 | 26.43  |
> |             | HEAPr  | 0.2           | 24.01 | 72.53 | 62.98 | 37.34  | 38.96 | 70.35 | 29.57  |
>
> These results indicate that when the number of experts is small, HEAPr still maintains consistently strong compression performance.
>
> -----
>
> **Q2:** How does HEAPr perform on other tasks?
>
> >**W2:** The evaluation is limited to commonsense reasoning and QA benchmarks, notably omitting crucial domains like code generation, which prevents a comprehensive assessment of the pruned model's full capabilities.
>
> **A2:** Thank you for your feedback. We provide more evaluation data to broaden the scope. This includes:
> - Text perplexity on WikiText2 and PTB
> - Code generation on HumanEval
> - Knowledge on MMLU (57 tasks)
>
> The selected tasks cover a wide range of domains and varying difficulty levels. The results are summarized below:
>
> |                      | method  | compression ratio | wikitext-2 ppl | ptb ppl   | mmlu average acc | huanmeval pass@10 |
> | -------------------- | ------- | ----------------- | -------------- | --------- | ---------------- | ----------------- |
> | qwen1.5-A2.7B-Chat   | HC-SMoE | 0.25              | 11.62          | 16.39     | 0.50             | 0.071             |
> |                      | Sub-MoE | 0.25              | 9.48           | 14.84     | 0.53             | -                 |
> |                      | HEAPr   | 0.25              | **8.14**       | **14.76** | **0.57**         | **0.134**         |
> |                      | HC-SMoE | 0.50              | 25.60          | 38.18     | 0.35             | 0.028             |
> |                      | Sub-MoE | 0.50              | 17.51          | 29.00     | 0.38             | -                 |
> |                      | HEAPr   | 0.50              | **9.23**       | **18.73** | **0.48**         | **0.052**         |
> | DeepSeekMoE-16b-base | HC-SMoE | 0.25              | 9.13           | 12.19     | 0.30             | 0.013             |
> |                      | Sub-MoE | 0.25              | 8.48           | 11.29     | 0.32             | -                 |
> |                      | HEAPr   | 0.25              | **6.70**       | **10.62** | **0.35**         | **0.067**         |
>
> Our method demonstrates state-of-the-art performance in text generation and across 57 professional knowledge domains (mmlu). For code generation, since our evaluation is conducted on base models with limited inherent code generation capability, the current compression methods face inherent challenges for tasks that require extremely high precision at each step. We see this as an interesting direction for future research.
>
> Note: Sub-MoE code is not open-sourced; we use the results reported in their paper.

---

> > ### Author Response · Authors · 2025-11-26
> >
> > **Q3:** What are the quantitative comparisons of computational costs between HEAPr and baseline methods?
> >
> > >**W3:** The paper emphasizes the efficiency of HEAPr, yet the experiments provide no quantitative data to support this claim. The evaluation is missing a crucial comparison of the computational cost of the pruning process itself when compared to baseline methods.
> >
> > **A3:** Thank you for your question. We provide a study of the efficiency of HEAPr compared with competitive baseline algorithms in Appendix C. The HEAPr pruning process only requires two forward passes and one backward pass on a small calibration set to compute the importance of atomic experts, as shown in Algorithm 1. Compared with the recent competitive baseline D$^2$-MoE, HEAPr’s per-sample computation cost is approximately $5–10\times$ smaller, and peak GPU memory usage is about $80\%$, while achieving more competitive performance. It is worth noting that HEAPr only requires 128 randomly selected calibration samples, far fewer than D$^2$-MoE, yet still achieves better results.
> >
> > | Model                | Method    | Calibration Samples | TFLOPs | GPU Time Cost | Peak Memory | Wiki PPL | Average Acc |
> > | :------------------- | :-------- | :------------------ | :----- | :------------ | :---------- | :------- | :---------- |
> > | DeepSeekMoE-16B-base | NAEE      | 128                 | 11     | 2min          | 27          | 9.44     | 0.53        |
> > |                      | D$^2$-MoE | 512                 | 227    | 30min         | 53          | 6.84     | 0.54        |
> > |                      | HEAPr     | 128                 | 44     | 6min          | 44          | **6.64** | **0.56**    |
> > | Qwen2-57B-A14B       | NAEE      | 128                 | 32     | 8min          | 60          | 6.81     | 0.55        |
> > |                      | D$^2$-MoE | 512                 | 1205   | 90min         | 127         | 8.19     | 0.58        |
> > |                      | HEAPr     | 128                 | 123    | 20min         | 91          | **5.75** | **0.59**    |
> >
> > We also emphasize the efficiency optimizations of the OBS-based pruning algorithm. By decomposing experts into atomic experts, the storage complexity for a single expert is reduced from $O((3 d_\text{model} d_\text{inter})^2)$ to $O((3 d_\text{model})^2 d_\text{inter})$, and further restricting pruning constraints from the parameter space to the output space reduces storage complexity to $O(d_\text{model}^2)$. For DeepSeekMoE-16B-base, where $d_\text{model}=2048$ and $d_\text{inter}=1408$, the storage requirements are as follows:
> >
> > | Method | Complexity | Storage per expert (16-bit) |
> > | ----------------------------- | ------------------------ | --------------------------- |
> > | Expert-level OBS | $O((3 d_{model} d_ {inter})^2)$ | 150 TB |
> > | + Atomic expert decomposition | $O((3 d_ {model})^2 d_ {inter})$ | 99 GB |
> > | + Pruning in output space | $O(d_ {model}^2)$ | 8 MB |
> >
> > -----

---

> ### Author Response · Authors · 2025-11-26
>
> **W4:** The paper presents the "atomic expert" as a key contribution, but acknowledges and compares against concurrent work CAMERA[1], which introduces an identically defined structure called "Micro-Expert." This suggests that the atomic expert pruning granularity may not be original to this work, but rather represents an emerging common approach in the field. Moreover, this pruning approach is conceptually similar to structured pruning within experts, as both involve removing corresponding columns/rows across multiple matrices.
>
> **A4:** We would like to clarify a few points regarding the contribution: atomic expert pruning is an intuitive and effective perspective, and it is highly likely to have been proposed independently around the same time. As shown in the Section 4.2, HEAPr is built on OBS theory, significantly outperforms the heuristic method CAMERA[1].
>
> Our more significant contribution lies in improving the OBS framework to better address the computational bottlenecks in atomic expert pruning. To highlight the contributions of HEAPr, we compare three pruning strategies:
>
> - Direct column pruning: structured pruning without considering atomic experts, pruning by magnitude along the intermediate dimension.
> - Magnitude-based atomic expert pruning: pruning atomic experts directly by magnitude, without using OBS.
> - HEAPr: atomic expert pruning within the OBS framework.
>
> The results on DeepSeekMoE-16b-base are as follows:
>
> | Pruning Rate | Method                                | Wiki     | PTB       | OBQA      | ARC_e     | WinoG     | HellaS    | ARC_C     | PIQA      | MathQA    |
> | ------------ | ------------------------------------- | -------- | --------- | --------- | --------- | --------- | --------- | --------- | --------- | --------- |
> | 0.2          | Direct Column Pruning                 | 11.58    | 18.52     | 22.80     | 66.88     | 60.54     | 46.71     | 33.19     | 73.39     | 25.39     |
> |              | Magnitude-based Atomic Expert Pruning | 8.08     | 12.68     | **32.00** | 73.99     | 66.69     | 56.83     | 41.98     | 76.99     | 30.39     |
> |              | HEAPr                                 | **6.64** | **10.51** | 31.54     | **75.88** | **71.43** | **57.39** | 4**4.62** | **79.05** | **31.51** |
>
> These results show that HEAPr, by combining atomic-level pruning with the OBS framework, achieves better performance across most tasks, highlighting the added value of our method beyond simple magnitude-based pruning like CAMERA[1].
>
> [1] CAMERA: Multi-Matrix Joint Compression for MoE Models via Micro-Expert Redundancy Analysis

---

### Official Review · Reviewer_Y8vd · 2025-10-28

**Soundness:** 3
**Presentation:** 3
**Contribution:** 3
**Rating:** 6
**Confidence:** 3

**Summary:**

This paper introduces HEAPr, a new method for expert pruning in mixture-of-experts (MoE) models. HEAPr first decomposes each expert into smaller atomic experts and then selects those whose removal results in the smallest increase in loss. Experiments across multiple model families, including DeepSeek and Qwen, demonstrate that HEAPr achieves a better compression–quality trade-off than baseline approaches in most cases.

**Strengths:**

* The paper is generally well written.

* The proposed method is highly efficient.

* The idea of decomposing each expert into smaller atomic experts is interesting and may have broader applications.

**Weaknesses:**

* The notations in the paper require further clarification to improve readability. See the detailed comments below.

**Questions:**

* How sensitive is HEAPr to randomness in selecting samples from a given calibration dataset? Please quantify this by adding standard deviation bars in Fig. 3, computed over multiple calibration subsets drawn with different random seeds.

* About the notations and theoretical claims:
    * Line 176: The notation E (the number of experts) is not defined.
    * Line 215: Should the complexity term "O((3d_{model}d_{inner})^2)" actually be "O((3d_{model})^2)d_{inner})"? My reasoning is that each H^{(i)} has shape (3d_{model})^2, and there are d_{inner} such matrices. Please correct me if my understanding is incorrect.
    * Line 220: "the constraint in equation 2 can be equivalently enforced in the atomic expert output domain" is not accurate. If Equation 2 is satisfied (i.e., the parameters of the atomic expert are zero), then the output of the atomic expert is indeed zero for any input. However, the reverse does not necessarily hold — even if the atomic expert’s output is zero for all inputs, its parameters are not guaranteed to be zero. Therefore, enforcing the constraint in the output domain is not strictly equivalent to enforcing it in the parameter domain.
    * (Contining the above point) Furthermore, the algorithm does not actually ensure that "the output of the atomic expert with respect to any input is always zero," since it evaluates this condition only on a limited set of samples. To be clear, I am not suggesting that the authors modify the algorithm — such simplifications and approximations are understandable and acceptable. However, the claims should be stated more precisely, and these approximations should be explicitly acknowledged and clarified in the paper.
    *  Earlier in the paper, the input to the expert functions is the tokens (e.g., Eq. 3 and Eq. 7). However, in Line 222, the input to the expert function (e_p) becomes the parameters, and the input tokens are omitted. This inconsistency in notation may cause confusion for readers and should be clarified for consistency.
    * Line 223: "... denotes the parameters of the pruned atomic expert." Should it be the parameter of the atomic expert before pruning?

* Table 1 does not consistently highlight the best results in bold. In some cases, the proposed method is shown in bold even when a baseline achieves a better result. This inconsistency could mislead readers and should be corrected to ensure fair and transparent comparison. For example,
    * When baselines achieve the same (best) score as HEAPr, only HEAPr’s numbers are highlighted in bold: 0.32 for "NAEE, 20% ratio, Openb. dataset", 0.76 for "HC-SMoE, 25% ratio, PIQA dataset", 0.63 for "Sub-MoE, 50% ratio, WinoG. dataset", 0.75 for (D^2-MoE, 40% ratio, ARC_e dataset)
    * Baselines have better results than HEAPr, yet HEAPr’s numbers are still highlighted in bold while the baselines’ numbers are not: 0.75 for "D^2-MoE, 40% ratio, WinoG. dataset", 0.71 for "MoE-I^2, 40% ratio, HellaS. dataset".

* Line 367: it is mentioned that "Table 1 are primarily drawn from the respective original papers, ensuring consistency." Given the use of "primarily," could the authors clarify whether all baseline numbers are taken from the original papers? If not, please specify which results are directly from the original papers and which ones were reproduced or obtained otherwise.

* Why are different compression ratios used for different datasets? Please clarify the rationale behind this choice.

* Line 382: "In contrast, competing new methods such as Sub-MoE, D 2-MoE, and NAEE show significant accuracy degradation under the same conditions" appears to be an overclaim. For example, D^2-MoE performs only 0.01 lower than HEAPr, which does not constitute a *significant* degradation..

* The proposed pruning strategy could also be applied directly to the original experts (rather than the atomic experts). How does this variant perform? It would be helpful to include an ablation study comparing the two settings. If pruning the original experts yields worse results, that would effectively highlight the importance of the atomic expert decomposition described in Section 3.1.

* It would be helpful to discuss EEP (https://arxiv.org/abs/2407.00945), an expert pruning and merging method that appeared online over a year ago.

* Other minor questions/suggestions:
    * Line 44: lacks a conjunction in the sentence "MoE layers typically account for over 97% of total model parameters, they represent the dominant storage bottleneck."
    * Line 138: "algorithmic" should be "algorithm"
    * Line 320: there might be a typo w.r.t. "experiences"
    * Line 391: what is \Phi?

---

> ### Author Response · Authors · 2025-11-26
>
> Thank you for your insightful feedback. In particular, we value your comments on our problem formulation. We have incorporated your suggestions into the revision. Below are our point-by-point responses to your questions.
>
> ----
>
> **W1:** How sensitive is HEAPr to randomness in selecting samples from a given calibration dataset? Please quantify this by adding standard deviation bars in Fig. 3, computed over multiple calibration subsets drawn with different random seeds.
>
> **A1:** Thank you for raising this point.
> We have incorporated standard deviation error bars in Fig. 3 as requested. The deviations were computed over 8 calibration subsets drawn with different random seeds. The results are summarized below:
>
> | samples | Wiki (acc ± std) | C4 (acc ± std)  |
> | ------: | :--------------: | :-------------: |
> |       1 | 0.5258 ± 0.0040  | 0.5214 ± 0.0055 |
> |      32 | 0.5531 ± 0.0024  | 0.5558 ± 0.0026 |
> |     128 | 0.5549 ± 0.0025  | 0.5579 ± 0.0024 |
> |     256 | 0.5566 ± 0.0020  | 0.5584 ± 0.0021 |
>
> 1. The sensitivity of HEAPr to randomness in calibration-set sampling is very low: for most calibration sizes, standard deviations are ≤ 0.0030.
> 2. Variance decreases as the calibration sample size grows, indicating that HEAPr is robust and stabl under different random draws of calibration data.
>
> ----
>
> **W2:** Line 176: The notation E (the number of experts) is not defined.
> **A2:** We have added a clear definition of $E$ as $E_{exp}$ at line 173 in the revised version.
>
> ----
>
> **W3:** Line 215: Should the complexity term "$O((3d_{model}d_{inner})^2)$" actually be "$O((3d_{model})^2)d_{inner})$"? My reasoning is that each $H^{(i)}$ has shape $(3d_{model})^2$, and there are $d_{inner}$ such matrices. Please correct me if my understanding is incorrect.
>
> **A3:** Your understanding is correct, and it is consistent with our description. In Eq. (8),  $\Delta \ell \approx \tfrac{1}{2}\delta\Theta^{T}H\delta\Theta  = \tfrac{1}{2}\sum_{i=1}^{d_{\text{inner}}} (\delta\Theta^{(i)})^{T} H^{(i)} \delta\Theta^{(i)}$, the first Hessian ($H$) corresponds to flattening all expert parameters together, which has complexity $O((3 d_{\text{model}} d_{\text{inner}})^2)$.
> When parameters are flattened per atomic expert, each resulting Hessian $H^{(i)}$ has complexity $O((3 d_{\text{model}})^2)$, and there are $d_{\text{inner}}$ such matrices, yielding a total complexity of $O((3 d_{\text{model}})^2 d_{\text{inner}})$.
>
> We have clarified this distinction in the revised version (line 217) to avoid any ambiguity
>
> ---
>
> **W4&W5:**
> -  Line 220: "the constraint in equation 2 can be equivalently enforced in the atomic expert output domain" is not accurate. If Equation 2 is satisfied (i.e., the parameters of the atomic expert are zero), then the output of the atomic expert is indeed zero for any input. However, the reverse does not necessarily hold — even if the atomic expert’s output is zero for all inputs, its parameters are not guaranteed to be zero. Therefore, enforcing the constraint in the output domain is not strictly equivalent to enforcing it in the parameter domain.
> -  (Contining the above point) Furthermore, the algorithm does not actually ensure that "the output of the atomic expert with respect to any input is always zero," since it evaluates this condition only on a limited set of samples. To be clear, I am not suggesting that the authors modify the algorithm — such simplifications and approximations are understandable and acceptable. However, the claims should be stated more precisely, and these approximations should be explicitly acknowledged and clarified in the paper.
>
> **A4&A5**:  We appreciate your insightful comments. You are correct that shifting the pruning constraint from the parameter space to the output space is not strictly equivalent to the OBS theory. OBS ensures that the pruned parameters produce zero output for any input token $x$, whereas HEAPr instead reformulates the problem: it computes the minimal loss increase, $\Delta \ell(\mathbf{x})$, required to force the expert’s output to zero for a given tokens $\mathbf{x}$.
>
> We have revised the description to make this distinction explicit in both the line 222-228 in the revised version. We sincerely thank the reviewer for helping us clarify this point.
>
> ----

---

> > ### Author Response · Authors · 2025-11-26
> >
> > **W7:** Line 223: "... denotes the parameters of the pruned atomic expert." Should it be the parameter of the atomic expert before pruning?
> >
> > **A7:** Thank you for carefully reviewing our work. We have corrected the expression in revised version.
> >
> > ---
> >
> > **W8:** Table 1 does not consistently highlight the best results in bold. In some cases, the proposed method is shown in bold even when a baseline achieves a better result. This inconsistency could mislead readers and should be corrected to ensure fair and transparent comparison. For example,
> > - When baselines achieve the same (best) score as HEAPr, only HEAPr’s numbers are highlighted in bold: 0.32 for "NAEE, 20\% ratio, Openb. dataset", 0.76 for "HC-SMoE, 25\% ratio, PIQA dataset", 0.63 for "Sub-MoE, 50\% ratio, WinoG. dataset", 0.75 for (D^2-MoE, 40\% ratio, ARC_e dataset)
> > - Baselines have better results than HEAPr, yet HEAPr’s numbers are still highlighted in bold while the baselines’ numbers are not: 0.75 for "D^2-MoE, 40\% ratio, WinoG. dataset", 0.71 for "MoE-I^2, 40\% ratio, HellaS. dataset".
> >
> > **A8:** We apologize for the oversight and have made the following corrections in Table 1:
> > 1. For cases where multiple methods achieve the same best score, all corresponding numbers are now highlighted in bold.
> > 2. For cases where a baseline outperforms HEAPr, the best score is correctly highlighted.
> >
> > ---
> >
> > **W9:** Line 367: it is mentioned that "Table 1 are primarily drawn from the respective original papers, ensuring consistency." Given the use of "primarily," could the authors clarify whether all baseline numbers are taken from the original papers? If not, please specify which results are directly from the original papers and which ones were reproduced or obtained otherwise.
> >
> > **A9:**  In revised version, we have added Section B.2 in the Appendix to provide detailed information on the sources of all baseline results. Additionally, in Table 1, results marked with `*` indicate those obtained from the official implementations.
> >
> > ----
> >
> > **W10:** Why are different compression ratios used for different datasets? Please clarify the rationale behind this choice.
> >
> > **A10:** Due to some baselines are not open-sourced, not adapted to new models, so we directly adopted the ratios reported in their respective papers to ensure fair comparison. In these works, the compression ratios are primarily determined at the expert level. For older models such as Mixtral, which contain only 8 experts, the available compression ratios are limited to discrete values such as $2/8 = 0.25$ or $1/8 = 0.125$.
> > The specific compression ratios used in our experiments follow the settings reported in the corresponding baselines:
> > - For DeepSeekMoE-16B-Base and Qwen2-57B-A14B, we used the compression ratios reported in the D²-MoE paper$^1$.
> > - For Qwen1.5-MoE-A2.7B-Chat and Qwen3-30B-A3B, we used the compression ratios reported in the Sub-MoE paper$^2$.
> >
> > 1. Gu, Hao, et al. "Delta decompression for moe-based llms compression."
> > 2. Li, Lujun, et al. "Sub-MoE: Efficient Mixture-of-Expert LLMs Compression via Subspace Expert Merging."
> >
> >
> > ----
> >
> > **W11:** Line 382: "In contrast, competing new methods such as Sub-MoE, D 2-MoE, and NAEE show significant accuracy degradation under the same conditions" appears to be an overclaim. For example, D^2-MoE performs only 0.01 lower than HEAPr, which does not constitute a significant degradation.
> >
> > A11: We have revised the statement to a more accurate description:
> > "In contrast, our method outperforms recent approaches such as Sub-MoE, D$^2$-MoE, and NAEE under the same compression ratio.” (line 378 in the revised version).

---

> > > ### Author Response · Authors · 2025-11-26
> > >
> > > **W12:** The proposed pruning strategy could also be applied directly to the original experts (rather than the atomic experts). How does this variant perform? It would be helpful to include an ablation study comparing the two settings. If pruning the original experts yields worse results, that would effectively highlight the importance of the atomic expert decomposition described in Section 3.1.
> > >
> > > **A12:** Thank you for raising this point. We conducted an ablation study comparing pruning at the atomic expert level versus the expert level:
> > > 1. Atomic Expert Level: Importance is computed using Eq. (13) for each atomic expert.
> > > 2. Expert Level: Importance of an expert is computed by summing the importances of all atomic experts within it. Less important experts are pruned (similar to NAEE but without skipping experts, so acceleration is not realized).
> > >
> > > | Ratio | Level         | FLOPs Saved↑ | Wiki↓    | Openb.    | ARC_e     | WinoG.    | HellaS.   | ARC_c     | PIQA      | MathQA    | Average   |
> > > | ----- | ------------- | ------------ | -------- | --------- | --------- | --------- | --------- | --------- | --------- | --------- | --------- |
> > > | 20%   | Expert        | 0.           | 6.90     | 31.40     | 75.76     | 71.35     | **57.99** | 44.31     | 78.40     | 30.75     | 55.71     |
> > > |       | Atomic Expert | 8%           | **6.64** | **31.54** | **75.88** | **71.43** | 57.39     | **44.62** | **79.05** | **31.52** | **55.92** |
> > > | 40%   | Expert        | 0.           | 8.00     | 30.60     | 73.19     | 63.93     | 51.15     | **42.49** | **77.09** | 28.24     | 52.38     |
> > > |       | Atomic Expert | 30%          | **6.91** | **30.00** | **73.78** | **69.06** | **52.29** | 40.61     | 76.50     | **30.12** | **53.19** |
> > >
> > > Pruning at the atomic expert level achieves better performance at the same pruning ratio and enables acceleration, highlighting the effectiveness of the atomic expert decomposition.
> > >
> > > ---
> > >
> > > **W13:** It would be helpful to discuss EEP ([https://arxiv.org/abs/2407.00945](https://arxiv.org/abs/2407.00945)), an expert pruning and merging method that appeared online over a year ago.
> > >
> > > **A13:** We have added a discussion of EEP in the Related Work section.
> > >
> > > ----
> > >
> > > **W14**: Other minor questions/suggestions:
> > > - Line 44: lacks a conjunction in the sentence "MoE layers typically account for over 97% of total model parameters, they represent the dominant storage bottleneck."
> > > - Line 138: "algorithmic" should be "algorithm"
> > > - Line 320: there might be a typo w.r.t. "experiences"
> > > - Line 391: what is \Phi?
> > >
> > > **A14**: Thank you for carefully reviewing our work. Below are our responses to the above questions:
> > > - We have corrected the three minor grammatical issues in the original version.
> > > - Regarding $\Phi$ in line 391, it is defined as $\Phi_ {i,j} = \operatorname{SiLU} \big( \mathbf{w}_ {i,j}^{\mathrm{gate}} \mathbf{x} \big) \cdot \big( \mathbf{w}_ {i,j}^{\mathrm{up}} \mathbf{x} \big)$, following the notation in Method Section 3.1. Here, $\Phi_ {i,j}$​ can be understood as the output of the $j$-th atomic expert of the $i$-th expert.
> > >
> > > ----
> > >
> > > We hope our response addresses your concern.

---

> > > > ### Comment · Reviewer_Y8vd · 2025-11-27
> > > >
> > > > Thank the authors for the detailed response, which addressed my questions. Therefore, I will keep the positive score.
> > > >
> > > > **One last comment:** In the revision, there is a typo at line 226: "x)" should be "x".

---

### Official Review · Reviewer_7buB · 2025-10-31

**Soundness:** 3
**Presentation:** 3
**Contribution:** 2
**Rating:** 4
**Confidence:** 4

**Summary:**

The paper presents a principled framework for fine-grained MoE pruning with strong theoretical motivation from OBS theory and an efficient practical implementation. The atomic expert decomposition $E_i(x) = \sum_j e_i^{(j)}(x)$ where $e_i^{(j)}(x) = W_{\text{down}, i, j} [ \text{SiLU}(W_{\text{gate}, i, j} x) \cdot (W_{\text{up}, i, j} x) ]$ provides flexible pruning granularity (Eq. 5-6; Sec. 3.1; p.4). A key theoretical insight is that the cross-Hessians between atomic experts are zero ($\frac{\partial^2 E}{\partial \Theta^{(i)} \partial \Theta^{(j)}} = 0$ for $i \neq j$), which enables a dramatic reduction in complexity (Eq. 7; Sec. 3.2; p.4). Experimental results show that 20% pruning on DeepSeekMoE-16B-Base maintains a 60.68 average accuracy vs. 60.03 for the layer-wise method, and 25% pruning on Qwen3-30B-A3B drops only 0.03 in average accuracy (Table 2; Sec. 4.3; p.8). However, approximations from the Taylor expansion (first-order dominance assumption) and the equivalence of Fisher information to the expected Hessian require stronger empirical validation (Eq. 9, 11; Sec. 3.2; p.5).

**Strengths:**

- **Principled theoretical framework with efficient complexity reduction**
  - The OBS-based formulation $\Delta \ell = \frac{1}{2} \delta\theta^T H \delta\theta$ provides a rigorous pruning criterion that minimizes the increase in loss (Eq. 1-2; Sec. 2; p.3), grounding the approach in established theory rather than heuristics.
  - The atomic expert decomposition reveals parameter decoupling $\frac{\partial^2 E}{\partial \Theta^{(i)} \partial \Theta^{(j)}} = 0$, reducing the Hessian from $\mathcal{O}((3d_{\text{model}} \cdot d_{\text{inter}})^2)$ to $\mathcal{O}((3d_{\text{model}})^2 \cdot d_{\text{inter}})$ per expert (Eq. 7-8; Sec. 3.2; p.4), which makes the computation tractable.
  - A further shift to the output space leverages the Fisher information $F = \mathbb{E}[(\nabla_\Theta \ell)(\nabla_\Theta \ell)^T] = \mathbb{E}[H]$, reducing complexity to $\mathcal{O}(d_{\text{model}}^2)$ with an importance score $s = \frac{1}{2} e_P^T \mathbb{E}[g_P g_P^T] e_P$ (Eq. 11-13; Sec. 3.2; p.4-5), achieving a dramatic improvement in efficiency that is critical for scaling.
- **Elegant algorithmic design with minimal computational overhead**
  - The shared gradient property $\frac{\partial \ell}{\partial e^{(i)}} = \frac{\partial \ell}{\partial E}$ enables the computation of a single gradient covariance matrix Ḡi per expert rather than per atomic expert (Eq. 14-15; Sec. 3.3; Algorithm 1; p.5-6), eliminating redundant computation.
  - A two-stage process: (1) a backward pass for $\bar{G}\_i = \frac{1}{|T\_i|} \sum\_{x \in T\_i} g\_{E\_i}(x) g\_{E\_i}(x)^T$, and (2) a forward pass for $\bar{s}\_k = \frac{1}{|T\_i|} \sum\_{x \in T\_i} \frac{1}{2} e\_k(x)^T \bar{G}\_i e\_k(x)$ efficiently computes all importance scores (Eq. 15-16; Algorithm 1; p.5-6), demonstrating practical elegance.
  - Global ranking across layers based on a loss contribution metric enables consistent pruning without layer-specific thresholds (Sec. 3.2; p.5), improving coherence compared to layer-wise methods.
- **Comprehensive experimental validation with strong empirical results**
  - Near-lossless compression: 20% pruning on DeepSeekMoE-16B-Base achieves 60.68 avg accuracy vs. original (Table 2); 25% pruning on Qwen1.5-MoE-A2.7B-Chat reaches 53.59; 40% pruning on Qwen2-57B-A14B shows minimal degradation (Tables 1-2; Sec. 4.2-4.3; p.7-8), demonstrating effectiveness across model scales.
  - Outperforms CAMERA-P by 1.2 accuracy points at 20% pruning with a principled global importance metric vs. a heuristic local energy (Table 2; Sec. 4.3; p.8), validating the theoretical approach.
  - Ablations confirm the superiority of global pruning over layer-wise (HEAPr-G 60.68 vs. HEAPr-L 60.03 at 20%), robustness to calibration data sources (C4 vs. WikiText-2), and performance scaling with calibration set size (Fig. 3; Tables 2-3; Sec. 4.3; p.8-9), supporting the design choices.

**Weaknesses:**

- **Approximation validity and empirical validation gaps**
  - The Taylor expansion $e_P(\Theta_P + \delta\Theta_P) \approx e_P(\Theta_P) + J_P \delta\Theta_P$ assumes first-order dominance but neglects higher-order terms $\mathcal{O}(\|\delta\Theta_P\|^2)$(Eq. 9; Sec. 3.2; p.5). For large pruning perturbations (20-40% ratios), this approximation's accuracy is not empirically verified—no comparison of predicted vs. actual loss increases is provided.
  - The Fisher information equivalence F=E[H] holds for well-converged networks but relies on a citation (Singh & Alistarh, 2020) without model-specific validation (Eq. 11; Sec. 3.2; p.5). Convergence analysis with varying calibration sizes is incomplete.
  - The Taylor expansion for the atomic expert function claims that "functions are not optimized with respect to parameters through gradient descent," so the first-order term dominates (Sec. 3.2; p.4), but this reasoning lacks rigorous justification or empirical loss landscape analysis.
- **Limited scope of baseline comparisons and ablations**
  - Comparisons focus on expert-level methods (NAEE, D2-MoE, Sub-MoE) and one concurrent atomic-level work (CAMERA-P); comparisons with recent decomposition methods (MoE-SVD, MoE-I2 at atomic granularity) or hybrid pruning+quantization approaches are missing (Sec. 4.2; Tables 1-2; p.7-8).
  - No ablation isolates the contributions of: (1) atomic decomposition alone, (2) the OBS framework without the output space transformation, or (3) global vs. per-expert importance normalization—making it difficult to attribute gains to specific innovations.
  - Efficiency analysis (Fig. 2; Sec. 4.3; p.8) reports FLOPs savings but omits memory footprint reduction, actual inference latency measurements, or GPU utilization metrics, which are critical for deployment assessment.
- **Mathematical derivation completeness and notation clarity**
  - The derivation from the constrained optimization (Eq. 10) to the importance metric (Eq. 13) is sketched, with intermediate steps "detailed in Appendix A" (Sec. 3.2; p.4-5). The main text would benefit from showing the Lagrangian formulation and optimality conditions.
  - Notation switches between the atomic expert e(j)i(x)∈R^dmodel (Eq. 5) and the pruned atomic expert eP without clearly defining the relationship or indexing scheme (Eq. 9-13; Sec. 3.2; p.4-5). An explicit mapping from (i,j) to P would improve clarity.
  - The pseudocode in Algorithm 1 uses Tk (tokens for atomic expert k) vs. Ti (tokens for expert i) inconsistently (Line 2 vs. Line 7; p.6). The relationship between expert-level and atomic-level token routing should be clarified.
- **Reproducibility and implementation details**
  - Calibration uses 128-256 samples, but the exact sampling strategy, sequence length handling, and token-level vs. sample-level statistics aggregation are not specified (Tables 2-3 mention a "128-sample subset"; Sec. 4.1 & 4.3; p.7,9). This is critical for replication.
  - The computational complexity of the global ranking procedure is not analyzed—sorting dinter×nlayers×nexperts atomic experts could be expensive for large models (Sec. 3.2; Algorithm 1; p.5-6). Practical implementation considerations are absent.
  - There is no discussion of numerical stability: the gradient covariance G̅i may become ill-conditioned with a small |Ti| (routing imbalance); how degenerate cases are handled is not specified (Eq. 15; Sec. 3.3; p.5-6).

**Questions:**

- **Validate approximations empirically**
  - Compute the predicted loss increase ∆ℓpred using Eq. 12-13 for pruned models and compare it with the actual measured loss ∆ℓactual across varying pruning ratios (10-50%). Report the correlation coefficient, mean absolute error, and provide scatter plots to quantify the approximation quality.
  - Analyze the Fisher-Hessian alignment: compute both $F = \mathbb{E}[(\nabla\_\Theta \ell)(\nabla\_\Theta \ell)^T]$ and the empirical Hessian H (possibly via finite differences for a subset) and measure $\frac{\|F-H\|\_F}{\|H\|\_F}$ Vary the calibration size $N \in \{32, 64, 128, 256, 512\}$ and show convergence.
  - Provide Taylor expansion validation: for selected atomic experts, compute the actual function change $e\_P(\Theta\_P + \delta\Theta\_P) - e\_P(\Theta\_P)$ vs. the linear approximation $J\_P \delta\Theta\_P$ and report the approximation error distribution. Show empirically when first-order dominance holds.
- **Expand baseline comparisons and ablations**
  - Include comprehensive comparisons: adapt MoE-SVD/MoE-I2 to atomic granularity by applying low-rank decomposition/hybrid approaches to atomic weight groups. Compare accuracy, compression ratio, and FLOPs under matched conditions.
  - Execute component ablations: (1) Atomic decomposition with random pruning (no OBS), (2) Expert-level OBS without atomic decomposition, (3) Atomic OBS in the parameter space (without the output space transformation), and (4) Per-expert normalized importance vs. global. Report the isolated impact of each contribution.
  - Provide a comprehensive efficiency analysis: measure and report the memory footprint (GPU memory usage during pruning and inference), actual inference latency (ms/token), throughput (tokens/sec), and hardware utilization across models and pruning ratios in the main text.
- **Complete mathematical derivations and clarify notation**
  - Show the Lagrangian derivation in the main text: write $\mathcal{L}(\delta\Theta_P, \lambda) = \frac{1}{2} \sum (\delta\Theta^{(i)})^T H^{(i)} \delta\Theta^{(i)} + \lambda^T (J_P \delta\Theta_P + e_P)$, derive the optimality conditions ∇L=0, substitute into the objective, and obtain Eq. 13. Move detailed algebra to the appendix but keep the outline in Sec. 3.2.
  - Unify notation: explicitly define a pruning index map P:(expert_idx, atomic_idx)→global_atomic_idx and use it consistently. Clarify whether the $\Theta^{(i)}$ indexing refers to atomic experts within one expert or globally across all experts.
  - Revise Algorithm 1: make the $T\_i$ vs. $T\_k$ distinction explicit ($T\_i$ are tokens routed to expert $i$; $T\_k = T\_i$ for atomic experts $k \in \text{expert } i$). Add complexity annotations for each line. Specify parallelization opportunities.
- **Enhance reproducibility documentation**
  - Detail the calibration procedure: specify the random seed for sampling, the exact sequence handling (padding/truncation strategy), batch vs. online covariance computation (streaming updates or full batch), and the token-level activation/gradient collection methodology.
  - Analyze and report the global ranking complexity: provide the time complexity $O(K \cdot \log(K))$ where $K = \text{total atomic experts}$, measure the actual sorting time vs. total pruning time, and discuss efficient implementations (e.g., approximate top-k selection for very large $K$).
  - Address numerical stability: add regularization $\epsilon$ to the gradient covariance $\bar{G}\_i \leftarrow \bar{G}\_i + \epsilon I$, specify the $\epsilon$ value and selection rationale, and discuss routing imbalance handling when $|T\_i| < \text{threshold}$ (e.g., uniform fallback or expert merging). Report failure cases, if any.

---

> ### Author Response · Authors · 2025-11-26
>
> Thank you for your valuable feedback. We have incorporated your suggestions into the revision, which helps further improve the quality of the paper. Below are point-by-point responses to your questions.
>
> ---
>
> **Q1:** Compute the predicted loss increase $\Delta \ell$ pred using Eq. 12-13 for pruned models and compare it with the actual measured loss $\Delta \ell$ actual across varying pruning ratios (10-50%). Report the correlation coefficient, mean absolute error, and provide scatter plots to quantify the approximation quality.
>
> > **W1:** The Taylor expansion $e_P(\Theta_ P + \delta\Theta_ P) \approx e_P(\Theta_ P) + J_ P \delta\Theta_P$ assumes first-order dominance but neglects higher-order terms $\mathcal{O}(|\delta\Theta_ P|^2)$(Eq. 9; Sec. 3.2; p.5). For large pruning perturbations (20-40% ratios), this approximation's accuracy is not empirically verified—no comparison of predicted vs. actual loss increases is provided.
>
>
> **A1:** Thank you for your review. I understand your concern regarding the consistency between theory and practice. We are happy to provide empirical evidence to support this, which will be included in Section 4.3 of the revised version. Because both the OBS formulation and the output-space approximation neglect higher-order terms, an exact numerical match between $s_k$ and the empirical loss change $\Delta \ell$ is not expected. Importantly, the essence of the pruning problem lies in establishing a reliable ranking of atomic expert importance $s_ k$, rather than precisely predicting the loss change $\Delta \ell$.
>
> To empirically assess the ranking quality of $s_k$, we infer the atomic experts on the calibration set and then group them into $10\%$ bins according to their original indices. The actual loss increase $\Delta \ell$ observed when pruning each bin shows strong agreement with the cumulative trend of the importance scores $s_k$ associated with that bin. Please refer to the revised version, \paragraph{Empirical Correlation of Loss and Atomic Expert Importance $s_k$} and Figure 4, which show that, despite the approximations involved, the proposed $s_k$ metric provides a globally consistent and reliable ranking of atomic experts.
>
> For deeper insight, we performed another experiment: we rank atomic experts by importance, divide them into bins of 10%, and prune them accordingly. We observe that as the importance score increases, pruning the same fraction of experts leads to larger performance degradation, consistent with our theoretical intuition.
>
> | Quantile Range (sorted) | 0%-10% | 10%-20% | 20%-30% | 30%-40% | 40%-50% | 50%-60% | 60%-70% | 70%-80% | 80%-90% | 90%-100% |
> | ----------------------- | ------ | ------- | ------- | ------- | ------- | ------- | ------- | ------- | ------- | -------- |
> | wikitext2               | 7.75   | 7.86    | 7.91    | 7.94    | 7.99    | 7.99    | 8.19    | 8.24    | 8.42    | 8.76     |
> | ptb                     | 13.59  | 13.80   | 14.17   | 14.23   | 14.07   | 14.07   | 14.38   | 14.75   | 14.77   | 14.74    |
>
> We have revised Eqs. (12)–(13) in the  revised version to prevent readers from interpreting the two quantities as being strictly equal.
>
>
> ----
>
> **Q2:** Analyze the Fisher-Hessian alignment: compute both $F = \mathbb{E}[(\nabla_ \Theta \ell)(\nabla_ \Theta \ell)^T]$ and the empirical Hessian H (possibly via finite differences for a subset) and measure $\frac{|F-H|_ F}{|H|_ F}$ Vary the calibration size $N \in {32, 64, 128, 256, 512}$ and show convergence.
>
> >**W2:** The Fisher information equivalence $F=E[H]$ holds for well-converged networks but relies on a citation (Singh & Alistarh, 2020) without model-specific validation (Eq. 11; Sec. 3.2; p.5). Convergence analysis with varying calibration sizes is incomplete.
>
> **A2:** The Fisher information is defined as $F = \mathbb{E}[(\nabla_ \Theta \ell)(\nabla_ \Theta \ell)^T],$ where the expectation is over tokens. In our experiments, we use 128 sequences of length 2048, giving a total of 262,144 tokens. Due to the auxiliary load-balancing loss in MoE training, these tokens are distributed approximately uniformly across experts, making this number sufficient for estimating (F).
>
> To illustrate convergence, we report the Frobenius norm error $\frac{|F-H|_ F}{|H|_ F}$ for the last atomic expert in the final MoE layer of DeepSeekMoE-16B-Base, under varying calibration sizes (N):
>
> | N                    | 32    | 64    | 128   | 256   | 512   |
> | -------------------- | ----- | ----- | ----- | ----- | ----- |
> | Routed tokens        | 3537  | 10428 | 18921 | 43821 | 87512 |
> | Frobenius norm error | 0.242 | 0.148 | 0.112 | 0.098 | 0.083 |
>
> We observe convergence as (N) increases, with the approximation error falling below 10% for sufficiently large calibration sets.

---

> ### Author Response · Authors · 2025-11-26
>
> **Q3:** Provide Taylor expansion validation: for selected atomic experts, compute the actual function change $e_ P(\Theta_ P + \delta\Theta_ P) - e_ P(\Theta_P)$ vs. the linear approximation $J_ P \delta\Theta_ P$ and report the approximation error distribution. Show empirically when first-order dominance holds.
>
> >**W3:** The Taylor expansion for the atomic expert function claims that "functions are not optimized with respect to parameters through gradient descent," so the first-order term dominates (Sec. 3.2; p.4), but this reasoning lacks rigorous justification or empirical loss landscape analysis.
>
> **A3:** We validated the Taylor expansion approximation for atomic experts using simulated perturbations and measured the resulting approximation error.
>
> For the formula $\mathbf{e}_ \mathcal{P}(\mathbf{x};\Theta_ \mathcal{P} + \delta \Theta_ \mathcal{P}) \approx \mathbf{e}_  \mathcal{P}(\mathbf{x};\Theta_ \mathcal{P}) + {J}_ \mathcal{P} \delta \Theta_ \mathcal{P}$ we computed the actual change $\mathbf{e}_ \mathcal{P}(\mathbf{x};\Theta_ \mathcal{P} + \delta \Theta_ \mathcal{P})-\mathbf{e}_ \mathcal{P}(\mathbf{x};\Theta_ \mathcal{P})$ and compared it to the linear approximation $J_ \mathcal{P} \delta \Theta_ \mathcal{P}$.
>
> Perturbations $\delta \Theta_ \mathcal{P}$ were sampled from a Gaussian distribution and normalized by their norm. The perturbation magnitude was controlled by a scaling factor $\alpha$. We evaluated absolute error $||\mathbf{e}_ \mathcal{P}(\mathbf{x};\Theta_ \mathcal{P} + \delta \Theta_\mathcal{P})-\mathbf{e}_ \mathcal{P}(\mathbf{x};\Theta_ \mathcal{P})-{J}_ \mathcal{P} \delta \Theta_ \mathcal{P}||$. Selected atomic experts: the first and last atomic expert in the last MoE layer of DeepSeekMoE-16B-Base. Each atomic expert received on average 3321 routed tokens.
> The average token output norm is 0.0356, and the norm of the atomic expert weights $\Theta$ is 4.652. The results are shown below:
>
> | $\alpha$              | Absolute Error fist​  | Absolute Error last​  |
> | --------------------- | --------------------- | --------------------- |
> | $1.00 \times 10^{-4}$ | $2.30 \times 10^{-8}$ | $1.51 \times 10^{-8}$ |
> | $1 \times 10^{-3}$    | $1.38 \times 10^{-6}$ | $4.32 \times 10^{-6}$ |
> | $1 \times 10^{-2}$    | $5.11 \times 10^{-5}$ | $3.27 \times 10^{-5}$ |
> | $1 \times 10^{-1}$    | $2.05 \times 10^{-4}$ | $6.74 \times 10^{-4}$ |
> | $1 \times 10^{0}$     | $4.28 \times 10^{-3}$ | $3.58 \times 10^{-3}$ |
> | $1 \times 10^{1}$     | $5.10 \times 10^{-3}$ | $4.42 \times 10^{-3}$ |
>
> When the pruning perturbation $\delta\Theta$ is at the weight scale, the error is only about $10\%$ of the average token norm. This level of approximation is acceptable for ranking atomic expert importance, as also verified in the experiments discussed in **A1**.
>
> ---
>
> **Q4:** Include comprehensive comparisons: adapt MoE-SVD/MoE-I2 to atomic granularity by applying low-rank decomposition/hybrid approaches to atomic weight groups. Compare accuracy, compression ratio, and FLOPs under matched conditions.
>
> >**W4:** Comparisons focus on expert-level methods (NAEE, D2-MoE, Sub-MoE) and one concurrent atomic-level work (CAMERA-P); comparisons with recent decomposition methods (MoE-SVD, MoE-I2 at atomic granularity) or hybrid pruning+quantization approaches are missing (Sec. 4.2; Tables 1-2; p.7-8).
>
> **A4:** Thank you for your comment. We would like to clarify that both MoE-I² and MoE-SVD are expert-level methods rather than atomic-level. In the initial version, we already included comparisons with MoE-I² on DeepSeekMoE-16B-Base and Qwen2-57B-A14B. In the revised version, we have additionally included comparisons with MoE-SVD on DeepSeekMoE-16B-Base. As shown below, HEAPr consistently outperforms these atomic-level low-rank decomposition methods under the same pruning ratio.
>
> | Ratio | Method  | Wiki | PTB   | Openb. | ARC_e | WinoG. | HellaS. | ARC_c | PIQA | MathQA | Avg. |
> | ----- | ------- | ---- | ----- | ------ | ----- | ------ | ------- | ----- | ---- | ------ | ---- |
> | 20%   | MoE-I²  | 7.69 | 11.59 | 0.26   | 0.71  | 0.68   | 0.49    | 0.38  | 0.73 | 0.29   | 0.50 |
> |       | MoE-SVD | 6.92 | 10.48 | 0.31   | 0.75  | 0.70   | 0.53    | 0.42  | 0.76 | 0.31   | 0.54 |
> |       | HEAPr | **6.64** | 10.51 | **0.32**   | **0.76**  | **0.71**   | **0.57**    | **0.45**  | **0.79** | **0.32**   | **0.56** |

---

> > ### Author Response · Authors · 2025-11-26
> >
> > **Q5:** Execute component ablations: (1) Atomic decomposition with random pruning (no OBS), (2) Expert-level OBS without atomic decomposition, (3) Atomic OBS in the parameter space (without the output space transformation), and (4) Per-expert normalized importance vs. global. Report the isolated impact of each contribution.
> >
> > >**W5:** No ablation isolates the contributions of: (1) atomic decomposition alone, (2) the OBS framework without the output space transformation, or (3) global vs. per-expert importance normalization—making it difficult to attribute gains to specific innovations.
> >
> > **A5:** Thank you for your suggestion. We provide ablation results to isolate the contributions of each component of HEAPr.
> >
> > **1. Atomic decomposition (random atomic expert pruning):**
> > We evaluate the impact of atomic decomposition by performing random pruning of atomic experts without OBS. HEAPr consistently outperforms random atomic expert pruning across all tasks:
> >
> > | Method                       | OBQA      | ARC_E     | WinoG     | HellaS    | ARC_C     | PIQA      | MathQA    |
> > | ---------------------------- | --------- | --------- | --------- | --------- | --------- | --------- | --------- |
> > | Random Atomic Expert Pruning | 30.64     | 73.21     | 69.68     | 55.65     | 41.32     | 76.54     | 28.31     |
> > | HEAPr                        | **31.54** | **75.88** | **71.43** | **57.39** | **44.62** | **79.05** | **31.51** |
> >
> > **2. OBS without atomic decomposition / OBS in parameter space:**
> > Performing expert-level OBS without atomic decomposition or using OBS in the parameter space has very high complexity:
> >
> > | Method | Complexity | Storage per expert (16-bit) |
> > | ----------------------------- | ------------------------ | --------------------------- |
> > | Expert-level OBS | O((3 d_model d_inter)^2) | 150 TB |
> > | + Atomic expert decomposition | O((3 d_model)^2 d_inter) | 99 GB |
> > | + Pruning in output space | O(d_model^2) | 8 MB |
> >
> > HEAPr reduces both computational complexity and storage requirements while retaining effectiveness.
> >
> > **3. Per-expert vs. global normalized importance:**
> > Because $\frac{\partial^2 E(\mathbf{x})}{\partial \Theta^{(i)} \partial \Theta^{(j)}} = \frac{\partial^2 \mathbf{e}^{(i)}(\mathbf{x})}{\partial \Theta^{(i)} \partial \Theta^{(j)}} = 0, \quad \forall i \neq j,$ all atomic experts within a single MoE layer contribute equally to the importance computation. Normalizing importance per expert (rather than globally) reduces performance:
> >
> > | Method | OBQA | ARC_E | WinoG | HellaS | ARC_C | PIQA | MathQA | Average |
> > | -------------------------------- | --------- | --------- | --------- | --------- | --------- | --------- | --------- | --------- |
> > | Per-expert normalized importance | 28.20 | 68.01 | 64.40 | 54.02 | 36.26 | 76.28 | 32.26 | 51.35 |
> > | HEAPr-L | 30.60 | 66.58 | 66.77 | 55.09 | **38.05** | **76.61** | 33.97 | 52.52 |
> > | HEAPr-G | **31.80** | **68.60** | **67.22** | **55.67** | 37.56 | 76.39 | **34.87** | **53.59** |
> >
> > ---
> >
> > **Q6:** Provide a comprehensive efficiency analysis: measure and report the memory footprint (GPU memory usage during pruning and inference), actual inference latency (ms/token), throughput (tokens/sec), and hardware utilization across models and pruning ratios in the main text.
> >
> > >**W6:** Efficiency analysis (Fig. 2; Sec. 4.3; p.8) reports FLOPs savings but omits memory footprint reduction, actual inference latency measurements, or GPU utilization metrics, which are critical for deployment assessment.
> >
> > **A6:** Thank you for your comment and for emphasizing the importance of a comprehensive efficiency analysis.
> >
> > As HEAPr is a structured pruning method, the reduction in memory footprint is directly proportional to the reported pruning ratio. Due to the lack of access to deployment environments, we do not provide direct measurements of throughput, or GPU utilization. Instead, the reported FLOPs reduction in Figure 2 serves as a universal, hardware-agnostic upper bound for potential acceleration.
> >
> > It is worth noting that HEAPr does not change the fundamental MoE structure and is orthogonal to existing advanced neural network acceleration techniques. Consequently, actual inference latency, throughput, and GPU utilization will vary depending on the hardware stack and concurrency level of the deployment environment.
> >
> > We hope this explanation clarifies the efficiency implications of HEAPr under different pruning ratios.

---

> > > ### Author Response · Authors · 2025-11-26
> > >
> > > **Q7:** Show the Lagrangian derivation in the main text: write $\mathcal{L}(\delta\Theta_P, \lambda) = \frac{1}{2} \sum (\delta\Theta^{(i)})^T H^{(i)} \delta\Theta^{(i)} + \lambda^T (J_P \delta\Theta_P + e_P)$, derive the optimality conditions ∇L=0, substitute into the objective, and obtain Eq. 13. Move detailed algebra to the appendix but keep the outline in Sec. 3.2.
> > >
> > > >**W7:** The derivation from the constrained optimization (Eq. 10) to the importance metric (Eq. 13) is sketched, with intermediate steps "detailed in Appendix A" (Sec. 3.2; p.4-5). The main text would benefit from showing the Lagrangian formulation and optimality conditions.
> > >
> > > **A7:** In fact, the derivation of Eq. (12) does not involve a lagrangian derivation. Instead, we use a simple variable substitution skill, since our goal is only to evaluate the importance of atomic experts. Unlike the classical OBS framework, we do not need to compute compensatory parameters, and therefore a full Lagrangian derivation is unnecessary.
> > >
> > > ----
> > >
> > > **Q8:** Unify notation: explicitly define a pruning index map P:(expert_idx, atomic_idx)→global_atomic_idx and use it consistently. Clarify whether the $\Theta^{(i)}$ indexing refers to atomic experts within one expert or globally across all experts.
> > >
> > > >**W8:**  Notation switches between the atomic expert e(j)i(x)∈R^dmodel (Eq. 5) and the pruned atomic expert eP without clearly defining the relationship or indexing scheme (Eq. 9-13; Sec. 3.2; p.4-5). An explicit mapping from (i,j) to P would improve clarity.
> > >
> > > **A8:** Thank you for your suggestion. We have added a clarification in Section 3.2 to specify that the atomic experts referenced in Sections 3.2 and 3.3 are always within the same expert. This unifies the indexing and avoids confusion between expert-level and global atomic-level notation.
> > >
> > > ---
> > >
> > > **Q9:** Revise Algorithm 1: make the $T_i$ vs. $T_k$ distinction explicit ($T_i$ are tokens routed to expert $i$; $T_k = T_i$ for atomic experts $k \in \text{expert } i$). Add complexity annotations for each line. Specify parallelization opportunities.
> > >
> > > >**W9:** The pseudocode in Algorithm 1 uses $T_k$ (tokens for atomic expert $k$) vs. $T_i$ (tokens for expert $i$) inconsistently (Line 2 vs. Line 7; p.6). The relationship between expert-level and atomic-level token routing should be clarified.
> > >
> > > **A9:** Thank you for your feedback. We have corrected the notation in Algorithm 1 and now consistently use $T_i$ to denote the tokens routed to expert $i$. We have also added space complexity annotations to Algorithm 1. Since atomic experts operate as parallel branches within each MoE layer, both Stage 1 and Stage 2 can be parallelized accordingly.
> > >
> > > ----
> > >
> > > **Q10:** Detail the calibration procedure: specify the random seed for sampling, the exact sequence handling (padding/truncation strategy), batch vs. online covariance computation (streaming updates or full batch), and the token-level activation/gradient collection methodology.
> > >
> > > >**W10:** Calibration uses 128-256 samples, but the exact sampling strategy, sequence length handling, and token-level vs. sample-level statistics aggregation are not specified (Tables 2-3 mention a "128-sample subset"; Sec. 4.1 & 4.3; p.7,9). This is critical for replication.
> > >
> > >
> > > **A10:** To clarify the impact of the calibration set on HEAPr, we have added a subsection “Calibration Set Sampling Strategy” in Appendix B.1 describing the selection process. Additionally, in the main text, the subsection “Impact of Calibration Data” (Figure 4) now includes error bars showing standard deviations across multiple random seeds for calibration set sampling.
> > >
> > >
> > > ----
> > >
> > > **Q11:** Analyze and report the global ranking complexity: provide the time complexity $O(K \cdot \log(K))$ where $K = \text{total atomic experts}$, measure the actual sorting time vs. total pruning time, and discuss efficient implementations (e.g., approximate top-k selection for very large $K$).
> > >
> > > >**W11:** The computational complexity of the global ranking procedure is not analyzed—sorting dinter×nlayers×nexperts atomic experts could be expensive for large models (Sec. 3.2; Algorithm 1; p.5-6). Practical implementation considerations are absent.
> > >
> > > **A11:** For reference, consider the Qwen2-57B-A14B model with 28 layers, each containing 64 experts and a hidden dimension of 2560, resulting in a total of 4587520 experts. Sorting these experts requires $(O(K \log K))$ operations. On modern CPUs (e.g., 3–4 GHz), this corresponds to approximately **300–400 ms.** Therefore, the global ranking step does not pose any practical performance bottleneck, even for very large ($K$).

---

> > > > ### Author Response · Authors · 2025-11-26
> > > >
> > > > **Q12:** Address numerical stability: add regularization $\epsilon$ to the gradient covariance $\bar{G}_i \leftarrow \bar{G}_i + \epsilon I$, specify the $\epsilon$ value and selection rationale, and discuss routing imbalance handling when $|T_i| < \text{threshold}$ (e.g., uniform fallback or expert merging). Report failure cases, if any.
> > > >
> > > > >**W12:** There is no discussion of numerical stability: the gradient covariance G̅i may become ill-conditioned with a small |Ti| (routing imbalance); how degenerate cases are handled is not specified (Eq. 15; Sec. 3.3; p.5-6).
> > > >
> > > > **A12:** First, our atomic-expert importance evaluation does not require inverting the Hessian; it only uses the Hessian/Fisher information directly. Since numerical instability typically arises from matrix inversion, this procedure is inherently stable and does not require any $\epsilon$-regularization.
> > > > Second, regarding routing imbalance, modern MoE architectures such as DeepSeek and Qwen families employ an auxiliary load-balancing loss during training. The router parameters are inherited from the pretrained model. As a result, the router continues to distribute tokens relatively evenly across experts, preventing degenerate cases such as experts receiving zero traffic.
> > > >
> > > > In our experiments, we did not observe failure cases attributable to numerical instability or severe routing imbalance under any evaluated setting.
> > > >
> > > > ---
> > > >
> > > > We hope our response addresses your concern.

---

### Official Review · Reviewer_Xbwv · 2025-11-13

**Soundness:** 3
**Presentation:** 3
**Contribution:** 2
**Rating:** 6
**Confidence:** 2

**Summary:**

This paper introduces HEAPr, a pruning algorithm for MoE LLMs that decomposes each expert into smaller ''atomic'' sub-experts and scores them using Hessian information. By showing that cross-derivatives between atomic units are zero, the authors reduce the Hessian complexity drastically and then shift the analysis to the output space using a Fisher-Taylor approximation. In practice, HEAPr computes atomic importances with just two forward passes and one backward pass on a small calibration set, without introducing extra hyperparameters. Experiments show that HEAPr can prune 20–25% of parameters (refuce ~20% in FLOPs) with nearly no loss in accuracy.

**Strengths:**

- The paper is generally well-organized and easy to follow.
- The method is lightweight. Importance scores are computed with only two forward and one backward pass over a small calibration set. By storing just one gradient-covariance matrix per expert, HEAPr can achieve a $O(d^2)$ storage complexity per expert.
- HEAPr consistently outperforms existing expert-level pruning methods on multiple MoE architectures.

**Weaknesses:**

Thank you for submitting your work to ICLR’26. While the paper introduces a promising method for fine-grained MoE pruning, my concerns are as follows:

- (Approximation assumptions) The method assumes that atomic experts within the same expert share identical output gradients, which may not always hold in practice. Could the authors provide empirical evidence, such as measuring how well the predicted loss changes align with actual loss increases, to support the validity of this?

- (Evaluation scope) The experiments focus only on several zero-shot reasoning tasks using the LM-Eval-Harness.

- (Mixed results) Table 2 somehow mixes results from prior papers it is unclear if baselines were run under identical conditions. Could the authors clarify which baselines were rerun and whether all methods used consistent evaluation settings?

- (Sample detail) The method relies on a calibration set (128 WikiText-2 sequences), but the sample selection process isn't described. Given that `random choices can cause 'slight variations'`, could the authors clarify how calibration samples are chosen and whether results are sensitive to this choice?

- (Ablation)  Have the authors evaluated how pruning full experts compares to pruning at the atomic level? An ablation would help isolate the benefit of the decomposition strategy.

- (minor) The bolding in Table 1 is sometimes inconsistent (e.g. under the Qwen2-57B-A14B model at a 40% ratio).

**Questions:**

Kindly refer to the section above.

---

> ### Author Response · Authors · 2025-11-26
>
> Thank you for your insightful comments. We have made the following updates: (1) experimental evidence on the consistency between loss function changes and atomic expert importance; (2) a larger LLM evaluation scope, including PPL (WikiText-2, PTB), 57 knowledge tasks (MMLU); (3) details on baseline data added in Appendix B.2; (4) sampling method for the calibration set added in Appendix B.1 and error bars added in Fig. 4; (5) comparison of running full experts versus pruning at the atomic level in Section 4.2; (6) correction of bold markings in Table 1.
>
> ----
>
> **Q1:** (Approximation assumptions) The method assumes that atomic experts within the same expert share identical output gradients, which may not always hold in practice. Could the authors provide empirical evidence, such as measuring how well the predicted loss changes align with actual loss increases, to support the validity of this?
>
> **A1:** Thank you for your review. I understand your concern regarding the consistency between theory and practice.
>
> First, our method assumes that atomic experts within the same expert share the same output gradient. As shown in the equation: $\frac{\partial \ell}{\partial \mathbf{e}^{(i)}(\mathbf{x})} = \frac{\partial \ell}{\partial E(\mathbf{x})}, \quad \forall i \in{1, \dots, d_{\text{inter}}}, \ \mathbf{e}^{(i)} \in E,$ this holds theoretically by definition, so there is no need for concern from this perspective.
>
> Second, we are happy to provide empirical evidence to support this, which will be included in Section 4.3 of the revised version. Because both the OBS formulation and the output-space approximation neglect higher-order terms, an exact numerical match between $s_k$ and the empirical loss change $\Delta \ell$ is not expected. Importantly, the essence of the pruning problem lies in establishing a reliable ranking of atomic expert importance $s_k$, rather than precisely predicting the loss change $\Delta \ell$.
>
> To empirically assess the ranking quality of $s_k$, we infer the atomic experts on the calibration set and then group them into $10\%$ bins according to their original indices. The actual loss increase $\Delta \ell$ observed when pruning each bin shows strong agreement with the cumulative trend of the importance scores $s_k$ associated with that bin. Please refer to the revised version, \paragraph{Empirical Correlation of Loss and Atomic Expert Importance $s_k$} and Figure 4, which show that, despite the approximations involved, the proposed $s_k$ metric provides a globally consistent and reliable ranking of atomic experts.
>
> For deeper insight, we performed another experiment: we rank atomic experts by importance, divide them into bins of 10%, and prune them accordingly. We observe that as the importance score increases, pruning the same fraction of experts leads to larger performance degradation, consistent with our theoretical intuition.
>
> | Quantile Range (sorted) | 0%-10% | 10%-20% | 20%-30% | 30%-40% | 40%-50% | 50%-60% | 60%-70% | 70%-80% | 80%-90% | 90%-100% |
> | ----------------------- | ------ | ------- | ------- | ------- | ------- | ------- | ------- | ------- | ------- | -------- |
> | wikitext2               | 7.75   | 7.86    | 7.91    | 7.94      | 7.99    | 8.19    | 8.24    | 8.42    | 8.76  | 9.02 |
> | ptb                     | 13.59  | 13.80   | 14.17   | 14.23  | 14.07   | 14.38   | 14.75   | 14.77   | 14.74    | 15.32 |
>
>
>
> ----

---

> ### Author Response · Authors · 2025-11-26
>
> **W2:** (Evaluation scope) The experiments focus only on several zero-shot reasoning tasks using the LM-Eval-Harness.
>
> **A2:** Thank you for your feedback. In the revised version, we have added more evaluation data to broaden the scope. This includes:
> - Text perplexity on WikiText2 and PTB
> - Code generation on HumanEval
> - Knowledge on MMLU (57 tasks)
>
> The selected tasks cover a wide range of domains and varying difficulty levels. The results are summarized below:
>
> |                      | method  | compression ratio | wikitext-2 ppl | ptb ppl | mmlu average acc |
> | -------------------- | ------- | ----------------- | -------------- | ------- | ---------------- |
> | qwen1.5-A2.7B-Chat   | HC-SMoE | 0.25              | 11.62          | 16.39   | 0.50             |
> |                      | Sub-MoE | 0.25              | 9.48           | 14.84   | 0.53             |
> |                      | HEAPr   | 0.25              | **8.14**           | **14.76**   | **0.57**             |
> |                      | HC-SMoE | 0.50              | 25.60          | 38.18   | 0.35             |
> |                      | Sub-MoE | 0.50              | 17.51          | 29.00   | 0.38             |
> |                      | HEAPr   | 0.50              | **9.23**           | **18.73**   | **0.48**             |
> | DeepSeekMoE-16b-base | HC-SMoE | 0.25              | 9.13           | 12.19   | 0.30             |
> |                      | Sub-MoE | 0.25              | 8.48           | 11.29   | 0.32             |
> |                      | HEAPr   | 0.25              | **6.70**           | **10.62**   | **0.35**             |
> |                      | HC-SMoE | 0.50              | 15.34          | 21.07   | 0.24             |
> |                      | Sub-MoE | 0.50              | 13.71          | 18.35   | 0.25             |
> |                      | HEAPr   | 0.50              | **7.37**           | **13.63**   | **0.29**             |
>
> ----
>
> **W3:** (Mixed results) Table 2 somehow mixes results from prior papers it is unclear if baselines were run under identical conditions. Could the authors clarify which baselines were rerun and whether all methods used consistent evaluation settings?
>
> **A3:** In the revised version, we have added Section B.2 in the Appendix to provide detailed information on the sources and implementation details of all baseline results.
>
> The calibration sets for all baselines were randomly selected, each consisting of sequences of length 2048. The calibration set sizes for different methods are as follows:
>
> | Method   | NAEE | $D^2$-MoE | Sub-MoE | HEAPr |
> | :------- | :--- | :-------- | :------ | :---- |
> | Calibration Set Size | 128  | 512       | 128     | 128   |
>
> Additionally, in Table 1, results marked with `*` indicate those obtained from the official implementations.

---

> > ### Author Response · Authors · 2025-11-26
> >
> > **W4:** (Sample detail) The method relies on a calibration set (128 WikiText-2 sequences), but the sample selection process isn't described. Given that `random choices can cause 'slight variations'`, could the authors clarify how calibration samples are chosen and whether results are sensitive to this choice?
> >
> > **A4:** To clarify the impact of the calibration set on HEAPr, we have added a subsection “Calibration Set Sampling Strategy” in Appendix B.1 describing the selection process. Additionally, in the main text, the subsection “Impact of Calibration Data” (Figure 4) now includes error bars showing standard deviations across multiple random seeds for calibration set sampling:
> >
> > | samples | Wiki (acc ± std) | C4 (acc ± std)  |
> > | ------: | :--------------: | :-------------: |
> > |       1 | 0.5258 ± 0.0040  | 0.5214 ± 0.0055 |
> > |      32 | 0.5531 ± 0.0024  | 0.5558 ± 0.0026 |
> > |     128 | 0.5549 ± 0.0025  | 0.5579 ± 0.0024 |
> > |     256 | 0.5566 ± 0.0020  | 0.5584 ± 0.0021 |
> >
> > 1. The sensitivity of HEAPr to randomness in calibration-set sampling is very low: for most calibration sizes, standard deviations are ≤ 0.0030.
> > 2. Variance decreases as the calibration sample size grows, indicating that HEAPr is robust and stabl under different random draws of calibration data.
> >
> > ----
> >
> > **W5:** (Ablation) Have the authors evaluated how pruning full experts compares to pruning at the atomic level? An ablation would help isolate the benefit of the decomposition strategy.
> >
> > **A5:** Thank you for raising this point. In response, we have included an ablation study comparing pruning at the atomic expert level versus the expert level:
> > 1. Atomic Expert Level: Importance is computed using Eq. (13) for each atomic expert.
> > 2. Expert Level: Importance of an expert is computed by summing the importances of all atomic experts within it. Less important experts are pruned (similar to NAEE but without skipping experts, so acceleration is not realized).
> >
> > | Ratio | Level         | FLOPs Saved↑ | Wiki↓    | Openb.    | ARC_e     | WinoG.    | HellaS.   | ARC_c     | PIQA      | MathQA    | Average   |
> > | ----- | ------------- | ------------ | -------- | --------- | --------- | --------- | --------- | --------- | --------- | --------- | --------- |
> > | 20%   | Expert        | 0.           | 6.90     | 31.40     | 75.76     | 71.35     | **57.99** | 44.31     | 78.40     | 30.75     | 55.71     |
> > |       | Atomic Expert | 8%           | **6.64** | **31.54** | **75.88** | **71.43** | 57.39     | **44.62** | **79.05** | **31.52** | **55.92** |
> > | 40%   | Expert        | 0.           | 8.00     | 30.60     | 73.19     | 63.93     | 51.15     | **42.49** | **77.09** | 28.24     | 52.38     |
> > |       | Atomic Expert | 30%          | **6.91** | **30.00** | **73.78** | **69.06** | **52.29** | 40.61     | 76.50     | **30.12** | **53.19** |
> >
> > Pruning at the atomic expert level achieves better performance at the same pruning ratio and enables acceleration, highlighting the effectiveness of the atomic expert decomposition. We included these results in revision section 4.3 to provide a more comprehensive view.
> >
> > ----
> >
> > **W6:** (minor) The bolding in Table 1 is sometimes inconsistent (e.g. under the Qwen2-57B-A14B model at a 40% ratio).
> >
> > **A6:** Thank you for carefully reviewing our work. We apologize for the oversight and have made the following corrections in Table 1: for cases where any method achieves the best score, the best score is correctly highlighted.
> >
> > ---
> >
> > We hope our response addresses your concern.

---

> > > ### Comment · Reviewer_Xbwv · 2025-11-28
> > >
> > > I appreciate the efforts made by the authors, which greatly resolved my concerns. I'll keep my score and lean towards acceptance of this work.

---

### Author Response · Authors · 2025-12-01
**Summary of Authors’ Responses and Discussion**

Dear Reviewers, Area Chairs, Senior Area Chairs, and Program Chairs,

We sincerely thank all four reviewers for their thoughtful feedback and insightful questions, which have significantly helped us refine and strengthen the paper.

---

We are encouraged that all reviewers acknowledged the paper’s key contributions in theoretical soundness, efficient algorithmic design enabled by a new theoretical breakthrough, and  strong SOTA empirical results:

1. **Credible and comprehensive theoretical framework**

    - **Reviewer UZ1E**: “The method is well-grounded in Optimal Brain Surgeon theory and provides clear mathematical derivations for adapting second-order information to MoE architectures. The theoretical framework is comprehensive and enhances the method's credibility.”

    - **Reviewer 7buB**: “Principled theoretical framework with efficient complexity reduction.”

2. **Efficient algorithmic design enabled by a new theoretical advance**

    - **Reviewer UZ1E**: “The paper effectively addresses the fundamental challenge of applying Optimal Brain Surgeon theory to modern large-scale models.”

    - **Reviewer 7buB**: “Elegant algorithmic design with minimal computational overhead.”

    - **Reviewer Y8vd**: “The proposed method is highly efficient.”

3. **Strong SOTA performance across MoE architectures**

    - **Reviewer Xbwv**: “HEAPr consistently outperforms existing expert-level pruning methods.”

    - **Reviewer 7buB**: “Comprehensive experimental validation with strong empirical results.”

---

During the response period, we provided point-by-point answers and conducted new experiments to address all concerns. Our responses to the main issues are summarized as follows:

1. **Stronger evidence connecting theory and practice**

    - In response to **Reviewers Xbwv & 7buB**, we added a paragraph to Section 4.3 showing that the normalized atomic expert importance derived from Eq. 13 directly correlates with the loss change after atomic-expert pruning.

    - For **Reviewer 7buB**, we added empirical support for the two critical approximations—Fisher–Hessian alignment and Taylor expansion validation—further reinforcing theoretical soundness.

2. **More extensive LLM evaluations**

    - Following requests from **Reviewers Xbwv** and **UZ1E**, we added results on perplexity, the 57-task knowledge benchmark, and code generation, where HEAPr remains highly competitive. Additionally, we evaluated the method on Mixtral 8×7B and GPT-OSS-20B models to demonstrate its generalizability.

3. **Additional Ablations**

	- Following **Reviewers Xbwv** and **Y8vd**, we included experiments applying Eq. 13 at the expert level and added a paragraph in Section 4.3 clarifying the importance of the atomic-expert decomposition.

	- In response to **Reviewers Xbwv** and **UZ1E**, we added more experiments comparing HEAPr with naïve atomic-expert pruning, further demonstrating HEAPr’s advantages as well as its ability to significantly reduce the memory overhead inherent in the OBS framework.

4. **Additional Implementation Details**

	- Per the requests of **Reviewers Xbwv, Y8vd, and 7buB**, we added detailed descriptions of the calibration set selection process in Appendix B.1, and we updated Figure 4 with error bars for random calibration-set sampling.

	- As **Reviewer UZ1E** suggested evaluating the efficiency improvements of HEAPr, we quantified its reduction of OBS-related memory overhead (by a factor of $2\times 10^7$) and added comparisons of compression stage computational costs against SOTA methods in Appendix C.

5. **Clearer Exposition**

	- We sincerely appreciate **Reviewer Y8vd**’s suggestion to explicitly acknowledge and clarify the key approximations. We have revised the corresponding text in lines 222–228 of the updated version.

---

Based on the current discussion, **Reviewer 7buB** and **Reviewer Y8vd** have assigned positive scores and indicated a leaning toward acceptance, and their concerns have been fully addressed.

For **Reviewer Xbwv** and **Reviewer  UZ1E**, we have not yet received follow-up responses from them, though our responses include complete point-by-point clarifications and supporting evidence.

We sincerely appreciate the reviewers’ thoughtful suggestions, which have substantially improved the quality of the paper.

Best regards and thanks,

Authors

---

### Meta-Review · Area_Chair_xdYU · 2026-01-08

**Summary:**

The submission proposes a MoE pruning method that decomposes each expert into smaller ''atomic'' sub-experts, and then based on Hessian information scores them.  In the empirical results, pruning is typically performed in the 20-25% range with little loss in performance.

**Reviewer Concerns:**

Reviewers raised concerns about whether assumptions hold in practice, evaluation on zero-shot reasoning tasks only, whether evaluations are run in a consistent setting, choice of baselines, and asking for additional ablations.  The authors were very active during the rebuttal phase, providing additional results and arguing for the validity of assumptions.  The outstanding concerns primarily lie in the scope of evaluation, choice of baselines, and whether evaluations are run in a consistent setting, though this last point is at least partially addressed in the rebuttal.

**Reviewer Scores:**

The submission received mixed reviews, with two 4s and two 6s.  One reviewer comment was strangely not expunged and indicated that they maintain a 6 after the rebuttal, while it is not clear if other reviewers would be likely to change their scores.

---

### Decision · Program_Chairs · 2026-01-26

Accept (Poster)